# Autophagosome membrane expansion is mediated by the N-terminus and *cis*-membrane association of human ATG8s

**Wenxin Zhang[1†], Taki Nishimura[1,2,3†], Deepanshi Gahlot[4,5], Chieko Saito[2], Colin Davis[6], Harold BJ Jefferies[1], Anne Schreiber[6], Lipi Thukral[4,5], Sharon A Tooze[1]\***

[1]Molecular Cell Biology of Autophagy Laboratory, The Francis Crick Institute, London, United Kingdom; [2]Department of Biochemistry and Molecular Biology, Graduate School and Faculty of Medicine, The University of Tokyo, Tokyo, Japan; [3]PRESTO, Japan Science and Technology Agency, Tokyo, Japan; [4]CSIR-Institute of Genomics and Integrative Biology, New Delhi, India; [5]Academy of Scientific and Innovative Research, Ghaziabad, India; [6]Cellular Degradation Systems Laboratory, The Francis Crick Institute, London, United Kingdom

**Abstract** Autophagy is an essential catabolic pathway which sequesters and engulfs cytosolic substrates via autophagosomes, unique double-membraned structures. ATG8 proteins are ubiquitin-like proteins recruited to autophagosome membranes by lipidation at the C-terminus. ATG8s recruit substrates, such as p62, and play an important role in mediating autophagosome membrane expansion. However, the precise function of lipidated ATG8 in expansion remains obscure. Using a real-time in vitro lipidation assay, we revealed that the N-termini of lipidated human ATG8s (LC3B and GABARAP) are highly dynamic and interact with the membrane. Moreover, atomistic MD simulation and FRET assays indicate that N-termini of LC3B and GABARAP associate in *cis* on the membrane. By using non-tagged GABARAPs, we show that GABARAP N-terminus and its *cis*-membrane insertion are crucial to regulate the size of autophagosomes in cells irrespectively of p62 degradation. Our study provides fundamental molecular insights into autophagosome membrane expansion, revealing the critical and unique function of lipidated ATG8.

## Editor's evaluation

In this study, the exciting possibility that Atg8s act on the membrane in "cis" is explored. While the correlation between in vitro data, MD simulations, and cell biology experiments could be further strengthened, the study presents a compelling case for giving serious consideration to the "cis" model.

## Introduction

Macroautophagy (hereafter autophagy) is a bulk intracellular degradation pathway, in which long-lived or damaged cytoplasmic materials are enveloped by double-membraned organelles, called autophagosomes, and delivered to lysosomes. Autophagy occurs in both basal and stressed conditions to maintain cellular homeostasis and cell survival (***Green and Levine, 2014***; ***Levine and Kroemer, 2008***). Upon autophagy induction, a small membrane structure, a phagophore, grows and expands, becoming a unique cup-shaped structure. Subsequently, the open edges of the phagophore close to form a spherical mature autophagosome (***Nakatogawa, 2020***). Autophagy is a highly dynamic

**\*For correspondence:**
sharon.tooze@crick.ac.uk

†These authors contributed equally to this work

**Competing interest:** The authors declare that no competing interests exist.

membrane process and sources lipids from multiple cellular membrane compartments, including ER, mitochondria, MAM (mitochondria-associated ER membrane), Golgi, ERES (ER exit sites), ERGIC (ER-Golgi intermediate compartment), and the plasma membrane (*Nishimura and Tooze, 2020*; *Tooze and Yoshimori, 2010*).

ATG8 is a unique ubiquitin-like protein conjugated to phospholipids on autophagic membranes (*Ichimura et al., 2000*; *Kabeya et al., 2004*). In yeast, there is only one Atg8, while in mammals there are at least six distinct ATG8 proteins, classified into two subfamilies, LC3s (LC3A, LC3B, and LC3C) and GABARAPs (GABARAP, GABARAPL1, and GABARAPL2) (*Mizushima, 2020*). These proteins are widely used as markers of autophagosomes. The conjugation reaction starts with the cleavage of ATG8 by ATG4 to expose a glycine residue at the C-terminus. Subsequently, ATG8 is activated by ATG7 (E1) and transferred to ATG3 (E2), and finally covalently conjugated to phosphatidylethanol-amine (PE) with the assistance of the ATG12-ATG5-ATG16L1 complex (hereafter, the E3 complex) (*Martens and Fracchiolla, 2020*). Autophagy cargo receptors, such as p62/SQSTM1, NBR1, and opti-neurin, are recruited to autophagic membranes by lipidated ATG8 proteins. These ATG8 interactors contain a consensus motif, called the Atg8-interacting motif (AIM) or the LC3-interacting region (LIR), which docks into two hydrophobic pockets on the surface of ATG8s (*Johansen and Lamark, 2020*; *Stolz et al., 2014*).

In addition to selectively binding cytosolic cargo, lipidated ATG8 proteins play an important role in mediating phagophore (also called isolation membrane) growth (*Xie et al., 2008*). In vivo evidence in cells suggests that ATG8 family proteins contribute to phagophore membrane expansion. Reduction in the protein level of Atg8 decreases the size of the autophagic bodies in yeast (*Xie et al., 2008*) and autophagosome size during *Caenorhabditis elegans* embryogenesis (*Wu et al., 2015*). In line with this, in mammalian cells, knock out of all ATG8 proteins results in smaller autophagosomes, though autophagosome formation is still maintained (*Nguyen et al., 2016*). Several in vitro studies have focused on the role of lipidated ATG8s in membrane tethering and fusion. The first studies employed in vitro reconstitution with yeast Atg7, Atg3, and Atg8 to study Atg8-PE and suggested Atg8-PE contributes to phagophore expansion by mediating hemifusion (*Nakatogawa et al., 2007*). Similar experiments have been employed to study mammalian ATG8 proteins using either ATG8s chemically conjugated to membranes or lipidated ATG8s to demonstrate the fusogenic properties of ATG8s (*Landajuela et al., 2016*; *Taniguchi et al., 2020*; *Weidberg et al., 2011*). These studies show association between lipidated ATG8 proteins on two distinct lipid bilayers ('in *trans*'), which proposed a model that phagophore membrane growth can be achieved by membrane tethering or fusion between vesicular compartments containing lipidated ATG8. In addition, a recent study demonstrated that two-aromatic membrane facing residues of Atg8 can associate with the membrane on which Atg8 is lipidated ('in *cis*') and induce positive membrane curvature, which suggest a poten-tial role of lipidated Atg8 in facilitating tubulovesicular structure formation (*Maruyama et al., 2021*). Despite these membrane association roles of lipidated ATG8, the underlying mechanism of ATG8-dependent phagophore expansion remains to be fully understood.

Crucially, unlike ubiquitin and other ubiquitin-like proteins, ATG8 family proteins contain two addi-tional α-helices at their N-termini, which is an evolutionary conserved feature (*Wesch et al., 2020*). Previous studies have reported several functions of the N-terminal regions, such as p62 recognition (*Shvets et al., 2008*; *Shvets et al., 2011*), membrane tethering (*Nakatogawa et al., 2007*; *Weidberg et al., 2011*; *Wu et al., 2015*), oligomerisation (*Coyle et al., 2002*; *Nakatogawa et al., 2007*), and direct lipid binding (*Chu et al., 2013*; *Sentelle et al., 2012*). These versatile roles might be achieved by the structural flexibility of ATG8 N-terminus (*Wu et al., 2015*).

Here we determine the conformation dynamics of LC3B and GABARAP N-terminal regions during lipidation using a number of approaches, including a real-time assay. We show that the N-termini of LC3B and GABARAP not only contribute to the interaction with ATG7 and ATG3 during the conjuga-tion reaction, but also associate with the membrane after their lipidation. Molecular dynamics (MD) simulation analyses suggest that the LC3B/GABARAP N-termini associate in *cis* on the membrane where lipidation occurs. The *cis*-membrane association was further confirmed using in vitro FRET assays. Furthermore, cells expressing non-tagged GABARAP mutants, lacking N-terminal regions or with impaired *cis*-membrane interactions of the N-terminus, form smaller autophagosomes, yet the degradation of p62 aggregation was fully restored. Our data elucidates a solo contribution of lipidated ATG8 N-terminus in phagophore membrane expansion without disrupting its cargo recognition. The

ATG8 N-terminus coordinates phagophore expansion and provides a molecular mechanism underlying the critical requirement of lipidated ATG8 in autophagy.

## Results

### N-terminal regions of LC3B/GABARAP are dynamically incorporated into hydrophobic environments during in vitro lipidation reaction

To investigate how the N-terminal regions of LC3B/GABARAP behave during the conjugation and lipidation reaction, we developed a real-time in vitro assay using LC3B/GABARAP proteins labelling with 7-nitrobenz-2-oxa-1,3-diazol-4-yl (NBD) (*Figure 1A*). As NBD is an environmentally sensitive fluorescence probe that displays enhanced fluorescence around 535 nm in a hydrophobic environment, and thus increases in NBD fluorescence reflect the location of LC3B/GABARAP N-termini in protein–protein and protein–membrane interfaces. LC3B/GABARAP were labelled with NBD by introducing single mutations at N-terminal residues: LC3B S3C$^{NBD}$ and GABARAP K2C$^{NBD}$, respectively. We first mixed ATG7, ATG3, the E3 complex, LC3B S3C$^{NBD}$, or GABARAP K2C$^{NBD}$ and liposomes, and after the addition of ATP, the increase in NBD signal was measured by fluorescence spectroscopy. As shown in *Figure 1B*, the fluorescence of LC3B S3C$^{NBD}$ was significantly increased in the presence of all components essential for efficient LC3 lipidation reaction ('All'). In contrast, no increase in fluorescence was observed without the E1 enzyme ATG7 ('no ATG7'), suggesting that the increase in NBD fluorescence is caused by the LC3 lipidation reaction. We observed a partial increase of the NBD fluorescence even in the absence of the E2 enzyme ATG3 ('no ATG3') or liposomes ('no liposomes') (*Figure 1B*), in which ATG7~LC3B and/or ATG3~LC3B intermediates, but not LC3B-PE, are formed. Furthermore, ATG7 and ATP were sufficient to induce a partial increase of NBD signals, which was enhanced by the addition of ATG3 (*Figure 1—figure supplement 1A and B*). Therefore, the NBD fluorescence increase reflects the formation of ATG7~LC3B and ATG3~LC3B intermediates as well as that of LC3B-PE (*Figure 1A*). We biochemically confirmed that LC3B S3C$^{NBD}$ was properly conjugated to ATG7, ATG3, and PE in in vitro reactions (*Figure 1C*). Similar results were acquired using GABARAP K2C$^{NBD}$ (*Figure 1D and E*, *Figure 1—figure supplement 1A and B*). These results suggest that the N-terminal regions of LC3B/GABARAP are incorporated into a hydrophobic environment during the conjugation (provided by the ATG7 or ATG3 interface) and lipidation reaction (provided by the membrane).

To dissect the effects of intermediate formation and lipidation on the increase in NBD fluorescence, we employed the real-time assay and added each component sequentially. The NBD signal of LC3B S3C$^{NBD}$ increased within 30 s after the addition of ATG7, independently of liposomes. After that, the E3 complex strongly stimulated the NBD signal increase in the presence of liposomes (*Figure 1F*, left). In the case of GABARAP K2C$^{NBD}$, similar responses of NBD signals were observed after the addition of ATG7, ATG3, and the E3 complex, while ATG3 more efficiently increased the NBD signal compared to ATG7 (*Figure 1F*, right). Thus, these results indicate that the N-termini of LC3B and GABARAP are highly dynamic during the lipidation reaction and encounter weak and strong hydrophobic environments during intermediate formation and lipid-conjugation, respectively.

We also assessed the lipidation activity of LC3B/GABARAP WT and their N-terminal deletion mutants LC3B ΔN11 and GABARAP ΔN9 by in vitro lipidation assays (*Figure 1—figure supplement 2*). Although in vitro ATG8 lipidation activity is more susceptible to the PE concentration in liposomes (*Nath et al., 2014*), in addition to high-amount PE containing liposomes ('lipidation-prone condition': 50% DOPE/50% POPC), we include a lipid composition, which mimics the PE level in the ER and contains negatively charged lipid phosphatidylinositol (PI) ('ER-like condition': 25% DOPE/10% Liver-PI/65% POPC). As expected, the lipidation of LC3B/GABARAP proteins is more efficient with the 'lipidation-prone' liposomes compared to the 'ER-like' liposomes (50% vs. 25% DOPE). Additionally, our data showed that ATG8 N-termini can be dispensable for their lipidation in either liposome condition, while the LC3B N-terminus is required for the full activity of in vitro lipidation reaction (*Figure 1—figure supplement 2*).

### Lipidated LC3B/GABARAP associate with membranes in *cis* analysed in silico

The real-time NBD assay demonstrated that the LC3B/GABARAP N-termini are located in hydrophobic surroundings after the lipidation reaction (*Figure 1F*), implying that the N-termini of lipidated

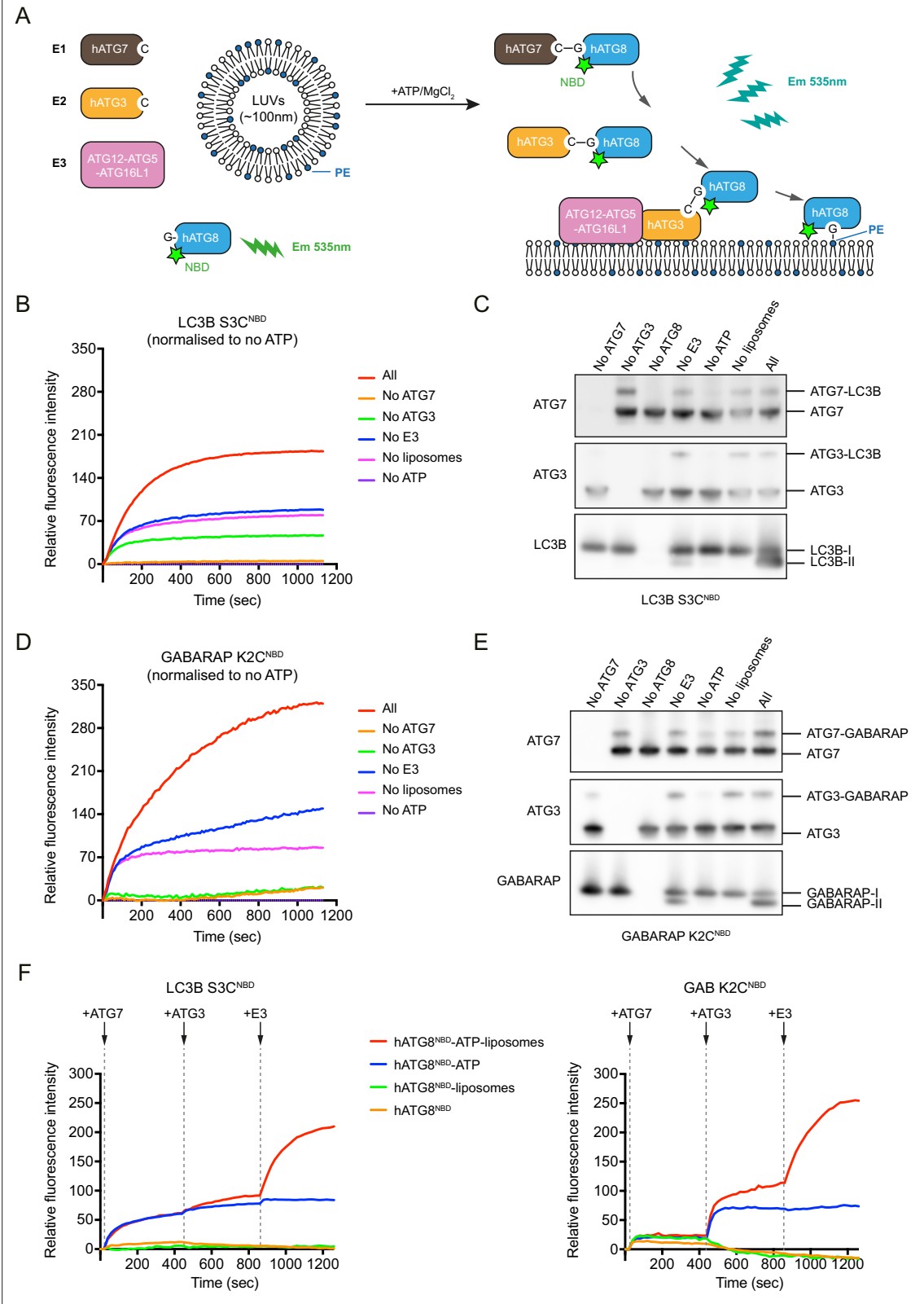

**Figure 1.** A real-time assay to track LC3B and GABARAP N-termini dynamics during lipidation. (**A**) A schematic diagram of the real-time lipidation assay using N-terminal NBD-labelled hATG8. (**B**) The NBD fluorescence changes of LC3B S3C$^{NBD}$ during lipidation reaction. 'All' condition contains 0.2 µM ATG7, 0.2 µM ATG3, 0.05 µM ATG12–ATG5-ATG16L1, 1 mM ATP, 1 mM large unilameller vesicles (LUVs) (50% DOPE/50% POPC) and 1 µM LC3B S3C$^{NBD}$, at 37°C. The NBD fluorescence was traced immediately once ATP was added (time point at 0 s). The rest conditions were performed in the absence of

*Figure 1 continued on next page*

*Figure 1 continued*

one component from the 'All' condition. The relative NBD fluorescence was normalised to 'no ATP' condition. Data represent mean values (n = 3). (**C**) Western blots of samples from (**B**). After 20 min reaction, the reactions from the real-time assay were stopped by adding sample buffer and immunoblot for ATG7, ATG3, and LC3B. (**D**) The NBD fluorescence changes of GABARAP K2C^NBD during lipidation reaction. Experimental conditions are the same as described in (**B**). Data represent mean values (n = 3). (**E**) Western blots of samples from (**D**). (**F**) Step-by-step real-time lipidation assay with LC3B S3C^NBD and GABARAP K2C^NBD. The NBD fluorescence was recorded continuously while adding ATG7 (between 60–80 s), adding ATG3 (between 480–500 s), and adding E3 complex (between 900–920 s). Data represent mean values (n = 3).

The online version of this article includes the following source data and figure supplement(s) for figure 1:

**Source data 1.** Uncropped blot and gel images of *Figure 1C and E* and source data file (Excel) for *Figure 1B, D and F*.

**Figure supplement 1.** The increase of LC3B/GABARAP N-terminal NBD fluorescence reflects the conjugation with ATG7, ATG3 and lipidation of LC3B/GABARAP.

**Figure supplement 1—source data 1.** Uncropped blot images of *Figure 1—figure supplement 1B* and source data file (Excel) for *Figure 1—figure supplement 1A*.

**Figure supplement 2.** LC3B N-terminus is required for in vitro lipidation reaction, while GABARAP N-terminus is dispensable for this process.

**Figure supplement 2—source data 1.** Uncropped gel images of *Figure 1—figure supplement 2A–D* and source data file (Excel) for *Figure 1—figure supplement 2*.

LC3B/GABARAP might be inserted into a lipid bilayer. So far, the structures of non-lipidated ATG8 proteins have been extensively studied by protein crystallisation techniques and NMR (*Sora et al., 2020*). These high-resolution structures have been employed in coarse-gained (CG) simulations which suggest that lipidated LC3B associates with membranes via a patch of basic residues (R68, R69, R70), β3, and the hydrophobic C-terminal region after β4 (*Fas et al., 2021*; *Thukral et al., 2015*; *Figure 1—figure supplement 1C*). On the other hand, there are two studies resolving the structure of lipidated GABARAP (*Ma et al., 2010*) and lipidated yeast Atg8 (*Maruyama et al., 2021*) on lipid bilayer nanodiscs using NMR. The NMR structures suggest similar membrane-associated regions in GABARAP and yeast Atg8, which are also demonstrated in the CG simulation studies. However, there are discrepancies between these studies which suggest different orientations of ATG8-PE on the membrane.

To further investigate the conformational rearrangements and structural dynamics of lipidated LC3B and GABARAP, we utilised all-atomistic MD simulations instead of static methods like molecular docking. Our structural analysis of membrane-embedded LC3B and GABARAP is based on six atomistic, explicit-water multiple MD simulation trajectories with a length of 1 µs each. The initial starting structures for these simulations were prepared after extensive orientation variability calculations on membrane to obtain unbiased position analysis (*Figure 2—figure supplement 1A*; see 'Materials and methods'). We found that the orientation variability calculations and membrane-facing residues converge to a unique and common conformation. Based on three replicas, the probability of each residue-lipid distance contact was computed and their occupancy, that is, presence during the simulation was calculated (*Figure 2B and D*).

In lipidated LC3B and GABARAP, simulations revealed four prominent membrane-interacting interfaces: the N-termini, loop 3, loop 6, and C-termini were consistently interacting with membrane albeit with dynamic movements of the side chains (*Figure 2*, *Figure 2—figure supplement 1B and C*). We observed that the N-termini of LC3B/GABARAP were inserted into the membrane over the length of the simulations. Compared to lipidated GABARAP, there are more membrane-associated residues in lipidated LC3B (*Figure 2B and D*). Unlike previous CG-MD simulations (*Fas et al., 2021*; *Thukral et al., 2015*), the basic residues in LC3B (R68, R69, R70, hereafter RRR) or GABARAP (R65, K66, R68, hereafter RKR) were distant from the membrane and facing towards cytosol in our atomistic models. Recently, the aromatic residues F77/F79 in lipidated yeast Atg8 were shown to be inserted into the membrane and regulate the membrane deformation in vitro (*Maruyama et al., 2021*). However, our atomistic MD simulation results show that the corresponding residues F80 in LC3B and F77/F79 GABARAP do not associate with the membrane, while additional membrane contacts are found in loop 6. Interestingly, our MD simulation suggests a *cis*-membrane association model of lipidated LC3B/GABARAP. The simulations (*Figure 2*, *Figure 2—figure supplement 1*) predict strong association of the N-termini and loops of LC3B/GABARAP with membrane lipids, suggesting that the N-termini and ubiquitin fold of LC3B-PE/GABARAP-PE interact on the same membrane (*Figure 3A and B*).

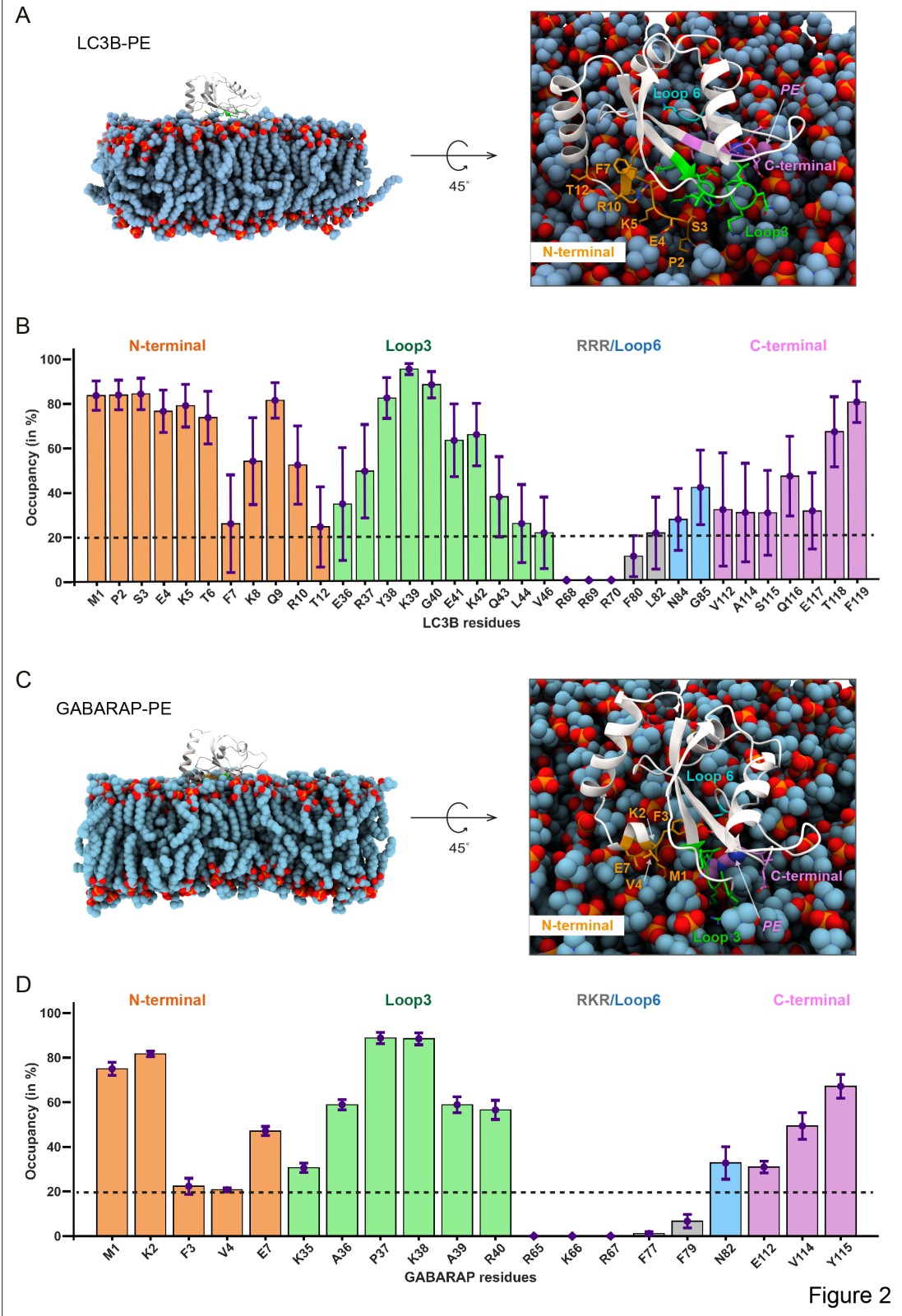

**Figure 2.** Molecular dynamics simulations of lipidated LC3B and GABARAP in POPC membrane elicit *cis*-membrane association of LC3B/GABARAP N-termini. (**A, C**) The representative structure of LC3B-PE (300 ns) and GABARAP-PE (840 ns). (**B, D**) The percentage of time (occupancy) LC3B/GABARAP residues are in contact with POPC membrane during 1 µs of the trajectory. The bars are coloured according to the region which is N-terminal as orange, loop 3 as green, loop 6 as light blue, and C-terminal as purple.

*Figure 2 continued on next page*

*Figure 2 continued*

The online version of this article includes the following figure supplement(s) for figure 2:

**Figure supplement 1.** Molecular dynamics (MD) simulation set-ups of lipidated LC3B and GABARAP in POPC membrane.

## N-terminal regions of liposome-conjugated LC3B/GABARAP are inserted into a lipid bilayer

To validate our model for membrane association of lipidated LC3B, we examined whether the N-terminus of membrane-conjugated LC3B is able to associate in cis. In brief, C-terminal $His_6$-tagged LC3B $S3C^{NBD}$ was mixed with liposomes containing nickel lipids and then the increases of NBD signals were analysed (*Figure 3C*). If the N-terminus of LC3B is close to membranes, thereby encountering a hydrophobic environment, the NBD signals are increased. When LC3B $S3C^{NBD}$-$His_6$ was mixed with POPC-based liposomes containing 5% Ni-NTA lipid, the fluorescence intensity of NBD was significantly increased compared to LC3B $S3C^{NBD}$-$His_6$ alone (*Figure 3D*). In addition, to investigate the effects of lipid saturation and PE on membrane association of LC3B N-terminus, we tested another three lipid compositions. These liposomes all contain 5% Ni-NTA-lipid, supplemented with 95% DOPC, 50% POPE/45% POPC, or 50% DOPE/45% POPC, respectively. As shown in *Figure 3D*, NBD signals increased in the presence of unsaturated lipids and/or PE, indicating that lipid-packing defects can facilitate membrane association of the LC3B N-terminus. We further investigated whether other regions of LC3B are involved in membrane association. K42 and L44 residues in loop 3 and R69 and R70 residues in RRR region are labelled with NBD. The fluorescence intensity of the NBD label from LC3B $K42C^{NBD}$-$His_6$ and LC3B $L44C^{NBD}$-$His_6$ was increased in the presence of liposomes containing nickel lipids, though the increase was less than half of the increase observed with LC3B $S3C^{NBD}$-$His_6$ (*Figure 3E*). In contrast, no fluorescence changes were detected with LC3B $R69C^{NBD}$-$His_6$ and LC3B $R70C^{NBD}$-$His_6$ (*Figure 3E*, *Figure 3—figure supplement 1A*). Consistent with the atomistic MD simulation, these results suggest that the N-terminal and loop 3 regions of lipidated LC3 are close to membranes compared to the RRR region. We performed the same assays with C-terminal $His_6$-tagged and NBD-labelled GABARAPs to examine the membrane association of GABARAP bound to liposomes. Consistent with our atomistic MD, the N-terminus ($K2C^{NBD}$) and loop 3 ($A39C^{NBD}$, $R40C^{NBD}$) of GABARAP associated with membranes (*Figure 3F*, *Figure 3—figure supplement 1B*). Surprisingly, in contrast to the MD simulation model, the RKR region ($R65C^{NBD}$, $K66C^{NBD}$) of GABARAP also showed NBD fluorescence increase.

To further validate that the increase of NBD fluorescence intensity results from protein-membrane association, we designed a FRET assay using NBD-labelled LC3B-$His_6$ or GABARAP-$His_6$ proteins and Rhodamine-labelled liposomes (*Figure 3G*). If the selected residues (*Figure 3A and B*) interact with liposomes containing rhodamine lipids, energy transfer should be detected, where the excitation of NBD results in an increase in rhodamine signal emission at 583 nm and a concomitant reduction of NBD signal at 535 nm (*Hui et al., 2011*). As shown in *Figure 3H*, we detected FRET between LC3B $S3C^{NBD}$-$His_6$ or GABARAP $K2C^{NBD}$-$His_6$ and rhodamine liposomes compared to experiments with blank liposomes without rhodamine lipids. In contrast, the increase in the NBD signal or FRET did not occur when LC3B $S3C^{NBD}$-$His_6$ or GABARAP $K2C^{NBD}$-$His_6$ was released with imidazole. FRET experiments were also performed with the other NBD-labelled LC3B-$His_6$ and GABARAP-$His_6$ proteins (*Figure 3—figure supplement 2*).

Our FRET experiments are consistent with the fluorescence-based liposome binding assays. The in vitro experimental results are in line with the atomistic MD simulation models, except for the GABARAP RKR region where we observed membrane association. One potential explanation is that the MD simulations model a single lipidated LC3B/GABARAP on a POPC membrane, whereas in the liposome-based assays, there might be multiple ATG8 proteins conjugated to one liposome, resulting conformational rearrangement of lipidated GABARAP (*Coyle et al., 2002*; *Wu et al., 2015*).

To confirm the *cis*-membrane association of ATG8 via its N-terminal residues, we introduced point mutations into the GABARAP N-terminus (Nmut: M1E/K2E/E7A) and labelled another membrane-associated residue ($V4C^{NBD}$, *Figure 2D*). We found that the increase in the NBD signal or FRET was reduced with GABARAP Nmut (*Figure 4A*, *Figure 4—figure supplement 1A*, *Figure 4—figure supplement 2A*), indicating a weakened *cis*-membrane association activity. In contrast, the membrane

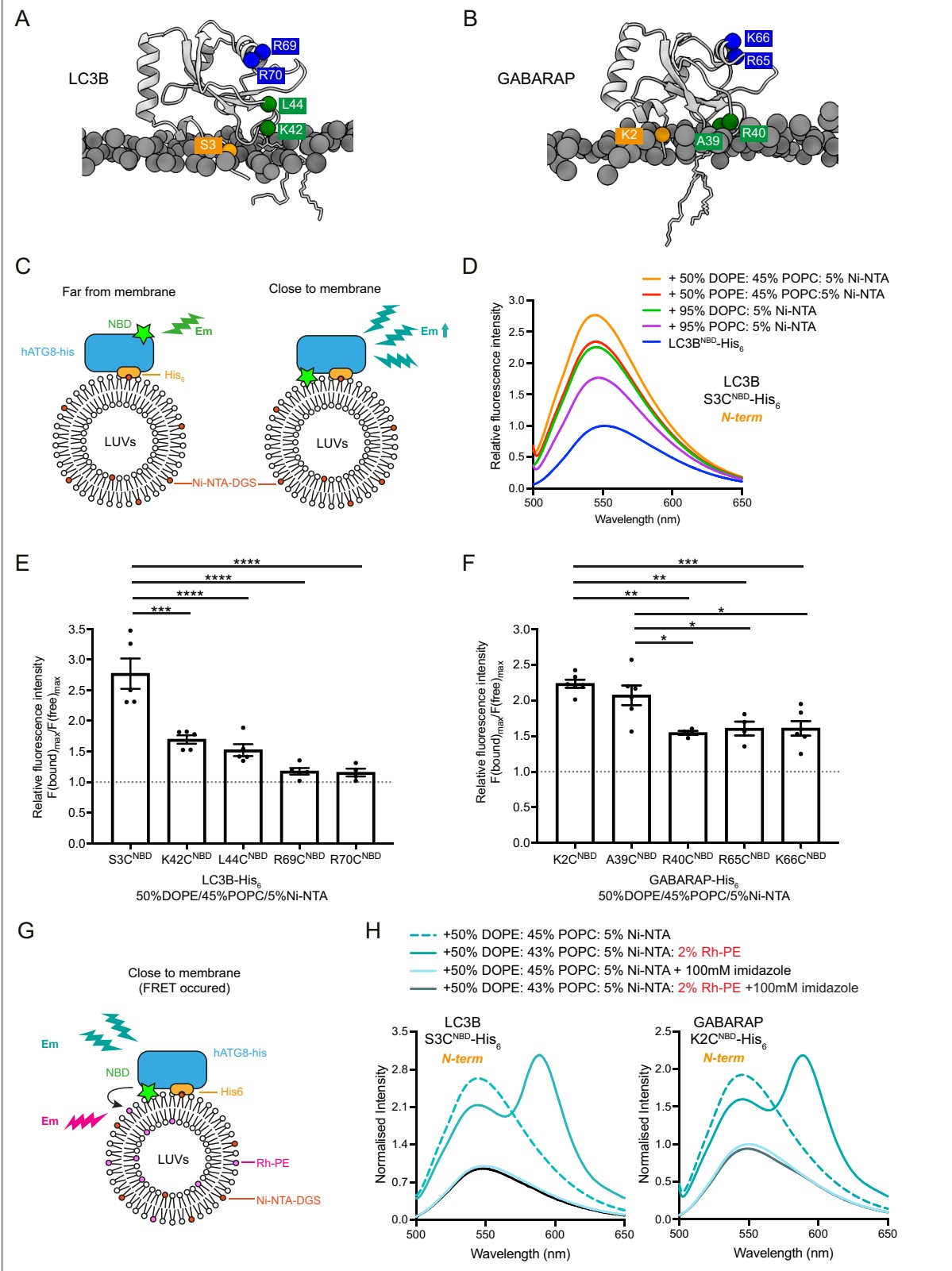

**Figure 3.** Analysis of membrane association interface of liposome-conjugated LC3B and GABARAP. (**A, B**) Representative structures of lipidated LC3B (300 ns) and GABARAP (840 ns) from Molecular dynamics (MD) simulation. The individual amino acids of interest are labelled with NBD. (**C**) Scheme for the NBD spectra assay to characterise the membrane association residues in LC3B and GABARAP, using LC3B-His$_6$/GABARAP- His$_6$ and large unilameller vesicles (LUVs) containing nickel lipids to mimic the lipidated status. If the residue is embedded in the membrane, the NBD fluorescence

*Figure 3 continued on next page*

*Figure 3 continued*

gets increased. (**D**) Example of fluorescence spectra of LC3B S3C$^{NBD}$- His$_6$ in the absence of LUVs (protein only, 1 µM) and in the presence of LUVs containing 5% Ni-NTA (1 mM). Spectra represent mean values (n = 5) (**E, F**) Quantification of NBD fluorescence increase of the screened residues in LC3B-His$_6$ and GABARAP-His$_6$. F(bound)$_{max}$/F(free)$_{max}$ ratio represents the maximum emission intensity of NBD-labelled LC3B-His$_6$/GABARAP-His$_6$ bound to liposomes containing 50% DOPE/45%POPC/5% Ni-NTA, normalised to the maximum emission intensity of NBD-labelled LC3B-His$_6$/GABARAP-His$_6$ in the absence of liposomes (n = 4–5, mean ± SEM). (**G**) Schematic diagram for the FRET assay to confirm the interaction between NBD-labelled LC3B-His$_6$/GABARAP-His$_6$ and rhodamine labelled liposomes. If the residue interacts with membrane, the emission of NBD would excite the rhodamine on the liposomes. (**H**) FRET assay with LC3B S3C$^{NBD}$-His$_6$ and GABARAP K2C$^{NBD}$-His$_6$. Each NBD-labelled LC3B-His$_6$/GABARAP-His$_6$ (1 µM) was mixed with 1 mM blank liposomes (50% DOPE/45% POPC/5% Ni-NTA) (dashed cyan) or rhodamine liposomes (50% DOPE/43%POPC/5% Ni-NTA/2% Rh-PE) (cyan). In parallel, addition of 100 mM imidazole to remove NBD-labelled LC3B-His$_6$/GABARAP-His$_6$ from liposomes was performed as negative controls. Spectra represent mean values (n = 3). Differences were statistically analysed by one-way ANOVA and Tukey multiple-comparison test. *p<0.05, **p<0.01, ***p<0.001, ****p<0.0001.

The online version of this article includes the following source data and figure supplement(s) for figure 3:

**Source data 1.** Source data file (Excel) for *Figure 3D, E, F and H*.

**Figure supplement 1.** Relative fluorescence spectra of NBD-labelled and liposome conjugated LC3B/GABARAP.

**Figure supplement 1—source data 1.** Source data file (Excel) for *Figure 3—figure supplement 1*.

**Figure supplement 2.** FRET between NBD-labelled LC3B-His$_6$/GABARAP-His$_6$ and rhodamine-labelled liposomes.

**Figure supplement 2—source data 1.** Source data file (Excel) for *Figure 3—figure supplement 2*.

tethering activity of GABARAP Nmut was maintained at the similar level to that of GABARAP WT (*Figure 4B and C*, *Figure 4—figure supplement 1B and C*, *Figure 4—figure supplement 2B and C*).

## Membrane insertion of LC3B/GABARAP N-termini is hindered by alteration of residues in loop 3 and RRR/RKR regions

As described above, we found that not only the N-termini, but also the loop 3 of LC3B/GABARAP associate with membranes in *cis* and that basic residues (RRR/RKR) are relatively detached from membranes after lipidation reaction. To further characterise the functions of the loop 3 and basic residues (RRR/RKR) in LC3B and GABARAP, we considered the apparent hydrophobicity of the residues in loop 3 (LC3B residues 42–44: KQL and GABARAP residues 39–41: ARI) and the RRR/RKR region and introduced triple glutamates (3E) which overall were either less hydrophobic or charge-inverting amino acids (*Wimley and White, 1996*). We employed N-terminal NBD-labelled LC3B or GABARAP as a probe and assessed the real-time lipidation activity of these mutants (*Figure 5A*). Note that no increases in NBD fluorescence were detected with LC3B S3C$^{NBD}$ KQL-3E (loop 3 mutant), while lipidation weakly occurred (*Figure 5B*). Moreover, for LC3B S3C$^{NBD}$ RRR-3E (RRR mutant), the conjugation reaction was completely blocked (*Figure 5C*). Thus, mutations in LC3B loop 3 and RRR regions hindered lipidation due to the impaired conjugation reaction with ATG7/ATG3 and/or N-terminal membrane association.

Distinct from LC3B loop 3 mutant, GABARAP K2C$^{NBD}$ loop 3 mutant (ARI-3E) was lipidated, while there was no increase in NBD fluorescence intensity (*Figure 5D*). The intermediates of GABARAP K2C$^{NBD}$ ARI-3E mutant with ATG7 and ATG3 can also be observed although to a lower level compared to GABARAP K2C$^{NBD}$ (*Figure 1E*). Thus, mutations in GABARAP loop 3 affected the conformation and hydrophobic environment of N-terminus during the lipidation reaction. With GABARAP K2C$^{NBD}$ RKR-3E, the NBD fluorescence intensities were gradually increased. Different from GABARAP K2C$^{NBD}$ (*Figure 1D*), the increased NBD signals indicated accumulations of ATG7~/ATG3~GABARAP intermediates (*Figure 5E*). GABARAP RKR mutant could not be discharged from ATG3~GABARAP intermediate, resulting in impaired GABARAP-PE conjugation.

## Selective degradation of p62 bodies does not require GABARAP N-terminus membrane insertion

We next investigated the effect of impaired membrane insertion of LC3B/GABARAP N-termini on autophagy in Hexa KO cells lacking all LC3/GABARAP subfamilies (*Nguyen et al., 2016*). Previous studies found that N-terminal GFP-tagged GABARAPs did not robust translocate to mitochondria upon mitophagy induction whereas the small HA-tag did, which suggests that the N-terminal fusion with large fluorescent tag may interfere with LC3B/GABARAP activity (*Lazarou et al., 2015*; *Nguyen*

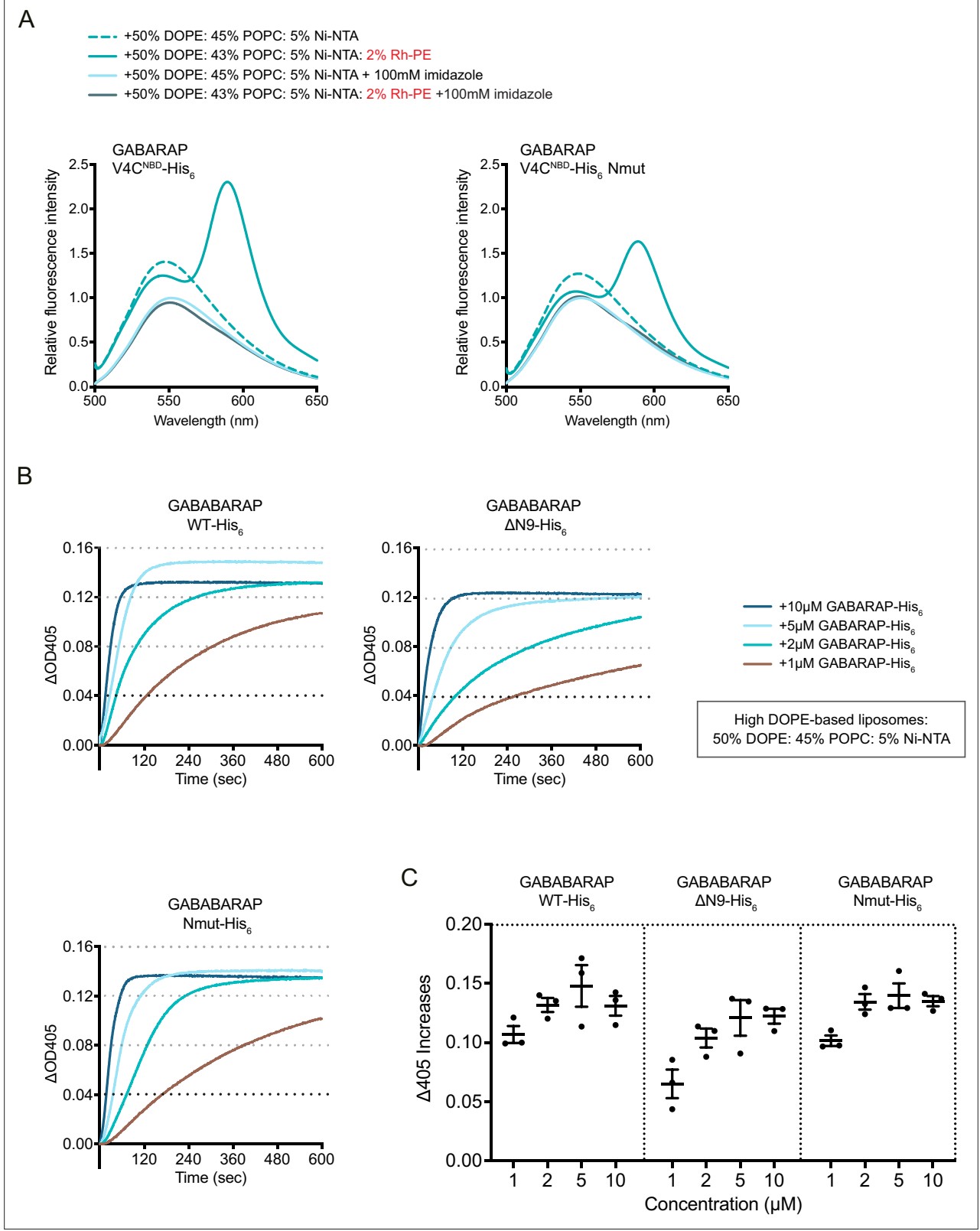

**Figure 4.** Point mutations in GABARAP N-terminus partially impair its N-terminal *cis*-membrane association but do not affect membrane tethering activity of GABARAP. (**A**) FRET assay with GABARAP V4C^NBD^-His$_6$ or V4C^NBD^ Nmut-His$_6$. Each NBD-labelled GABARAP-His$_6$ (1 μM) was mixed with 1 mM blank liposomes or rhodamine liposomes: 50% DOPE/45% POPC/5% Ni-NTA (dashed cyan) or 50% DOPE/43% POPC/5% Ni-NTA/2% Rh-PE (cyan). Addition of 100 mM imidazole to remove NBD-labelled LC3B-His$_6$/GABARAP-His$_6$ from liposomes was performed as negative controls. Spectra

*Figure 4 continued on next page*

*Figure 4 continued*

represent mean values (n = 3). (**B**) Liposome tethering assays with GABARAP-His$_6$, N-terminal deletion ΔN9-His$_6$ and Nmut-His$_6$ with 1 mM high DOPE-base liposomes containing 50% DOPE/45% POPC/5% Ni-NTA. Various amount of C-terminal His$_6$-tagged GABARAP proteins were used. Liposome tethering of each condition (ΔOD$_{405}$) was measured and normalised to that of liposomes only (n = 3). (**C**) Membrane tethering capacity after 10 min (ΔOD$_{405}$ increase) of each condition shown in (**B**) was plotted (n = 3, mean ± SEM).

The online version of this article includes the following source data and figure supplement(s) for figure 4:

**Source data 1.** Source data file (Excel) for *Figure 4A and B*.

**Figure supplement 1.** The same experiments have been done with POPC-based liposomes.

**Figure supplement 1—source data 1.** Source data file (Excel) for *Figure 4—figure supplement 1A and B*.

**Figure supplement 2.** The same experiments have been done with ER-like liposomes.

**Figure supplement 2—source data 1.** Source data file (Excel) for *Figure 4—figure supplement 2A and B*.

*et al., 2016*). To exclude potential artefacts caused by tagging ATG8, we used non-tagged GABARAP WT, an N-terminal deletion mutant (GABARAP Δ9), *cis*-membrane impaired mutants (GABARAP Nmut and ARI-3E), and the LIR-docking mutant (Y49A/L50A, YL-AA). Hexa KO cells stably expressing the non-tagged GABARAPs were cultured under fed or starved conditions with and without lysosomal inhibitor bafilomycin A$_1$ (BafA1) to monitor autophagic activity. GABARAP WT and all GABARAP mutants were efficiently lipidated (*Figure 6A and B*). Hexa KO cells showed an accumulation of high molecular weight forms (high-MW, *) of p62 at approximately 170 kDa, which represents covalently crosslinked p62 oligomers (*Donohue et al., 2014*). This high-MW p62 tended to be efficiently degraded in the cells expressing GABARAP WT and the mutants (*Figure 6A*, *Figure 6—figure supplement 1*). S349-phosphorylated p62, which reflects the gel-like state of p62 (*Kageyama et al., 2021*), was clearly detected in Hexa KO cells, but completely suppressed by the expression of GABARAP WT and mutants (*Figure 6A and C*). Consistent with the observed rescue of phospho-p62 degradation, the expression of non-tagged GABARAP WT and mutants significantly reduced the size of p62 bodies (*Figure 6D and E*). These results indicate that GABARAP N-terminus and *cis*-membrane insertion can be dispensable for the reduction of p62 bodies upon starvation. In contrast, the LIR-docking mutant (YL-AA), which does not interact with p62 (*Behrends et al., 2010*), was not functional in the degradation of high-MW p62 and p62 bodies (*Figure 6A–E*). Collectively, these results suggest that GABARAP N-terminus and *cis*-membrane insertion are not critical for p62 body degradation. Although we tried to address whether GABARAP mutants interact with ATG2 as well as GABARAP WT (*Bozic et al., 2020*), FLAG-ATG2A was not co-precipitated with GABARAP in our experimental conditions, which is probably due to its weak binding affinity (*Figure 6F*).

## Autophagosome membrane expansion requires GABARAP N-terminus and its membrane insertion

We next examined whether the N-terminus and/or *cis*-membrane insertion of GABARAP plays a role in the membrane expansion of autophagosomes. To analyse closed autophagosomes in the Hexa KO cells, we introduced GFP-tagged STX17 transmembrane domain (STX17TM: 229aa-275aa of STX17) and then expressed non-tagged GABARAPs. We characterised the ultrastructure of STX17TM-positive compartments in a region of interest (ROI) using correlative light and electron microscopy with 25-nm-thick serial sections (3D-CLEM). The size and volume of the autophagosomal structures that correlated with GFP-STX17TM signals were measured in each cell line under starved conditions (*Maeda et al., 2020*). The median of the major diameters of autophagosomal structures in Hexa KO cells was 290 nm, increasing significantly to 550 nm in the cells expressing GABARAP WT (*Figure 7A and B*). In contrast, *cis*-membrane insertion mutants (Nmut and ARI-3E) did not fully rescue the size of the autophagosomes (*Figure 7A and B*). In the Hexa KO cells expressing Nmut and ARI-3E, the median of the major diameters of autophagosomal structures was 380 nm and 410 nm, respectively. In these cells, the estimated volume autophagosomes was less than 42% of those observed in the cells expressing GABARAP WT (*Figure 7C*). In the case of N-terminal deletion mutant (Δ9), no restoration of autophagosome size was observed (*Figure 7A–C*). These results indicate that the N-terminal region of GABARAP plays a central role in autophagosome membrane expansion, which can be partly explained by its *cis*-membrane insertion. Given that the LIR-docking mutant (YL-AA) efficiently restored autophagosome size and volume to the same extent as GABARAP WT, we conclude that membrane

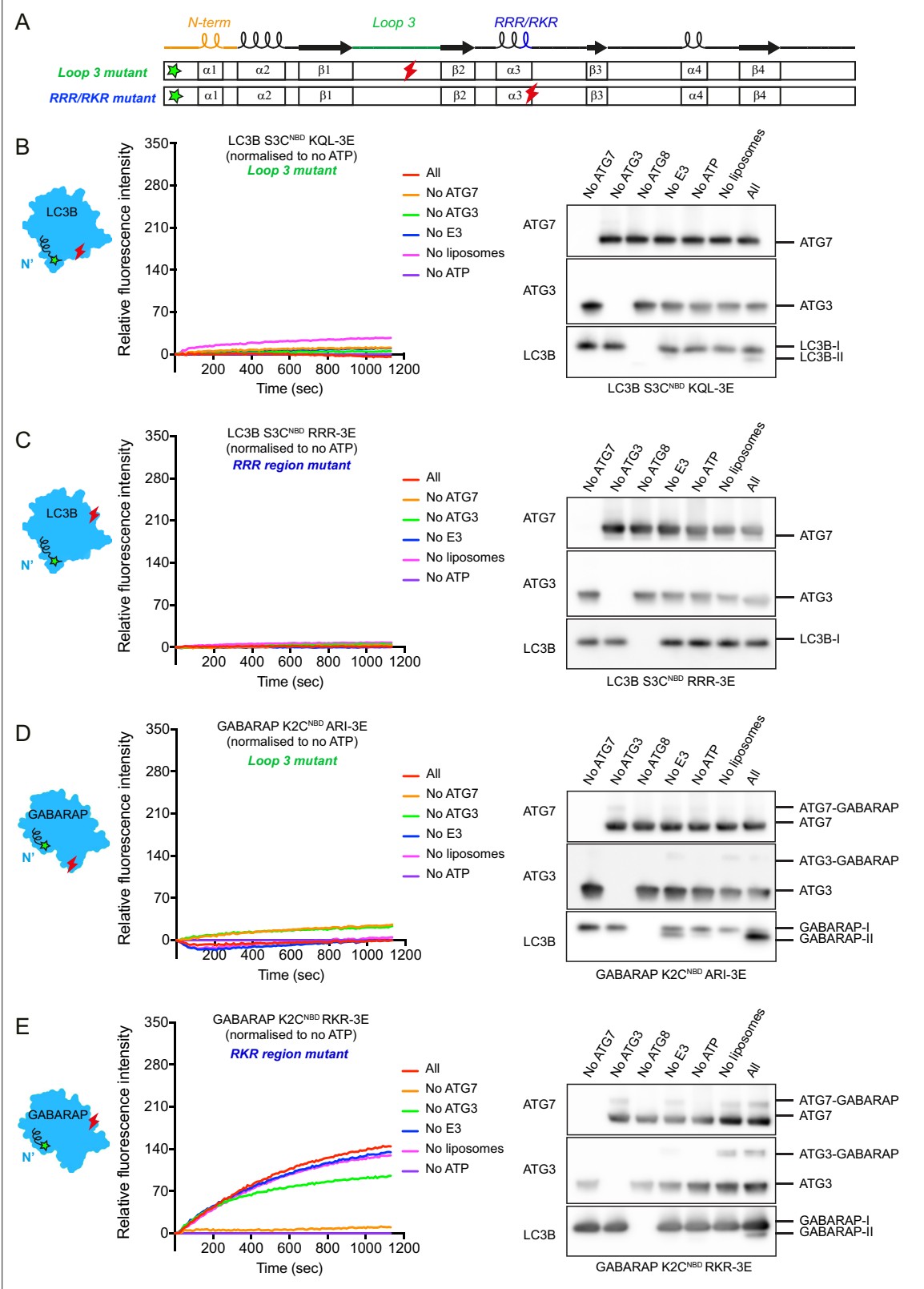

**Figure 5.** Membrane insertion of LC3B/GABARAP N-termini is hindered by alteration of residues in loop 3 and RRR/RKR regions. (**A**) Scheme of N-terminal NBD-labelled LC3B (S3C$^{NBD}$) or GABARAP (K2C$^{NBD}$) with less hydrophobic and charge-inverting mutations (3E) in loop 3 and RRR/RKR regions. The green and red marks indicate the position of NBD labelling and 3E mutations, respectively. (**B**) The real-time lipidation assay with LC3B S3C$^{NBD}$ KQL-3E (loop 3 mutant). (**C**) The real-time lipidation assay with LC3B S3C$^{NBD}$ RRR-3E (RRR region mutant). (**D**) The real-time lipidation assay with GABARAP

*Figure 5 continued on next page*

*Figure 5 continued*

K2C[NBD] ARI-3E (loop 3 mutant). (**E**) The real-time lipidation assay with GABARAP K2C[NBD] RKR-3E (RKR region mutant). All spectra represent mean value (n = 3).

The online version of this article includes the following source data for figure 5:

**Source data 1.** Uncropped blot images of *Figure 5B–E* and source data file (Excel) for *Figure 5B–E*.

expansion occurs independently of ATG8-cargo interaction under starved conditions. Taken together, these results support the conclusions that the GABARAP promotes phagophore expansion and mediates the autophagosome size through the N-terminus and its *cis*-membrane insertion.

## Discussion

The function of ATG8 proteins has been a long-standing question in autophagy research field. In this study, we have discovered the dynamic nature of ATG8 N-terminus and revealed how lipidated ATG8 regulates membrane expansion.

We developed an NBD fluorescence assay to monitor ATG8 lipidation reaction in vitro (*Figure 1*). The conjugation and lipidation cascades can be tracked by the incremental increase in the hydrophobic environment encountered by the N-termini of ATG8 proteins. These results clearly elucidate the highly dynamic nature of ATG8 N-termini during the lipidation reaction. Our NBD lipidation assay is capable of measuring ATG8 lipidation reaction on ~100 nm LUVs in real time and decipher the kinetics of the conjugation reactions. Excitingly, these approaches can be extended to further studies on the effects of upstream factors, for example, the role of ATG2-dependent lipid transport (*Maeda et al., 2019*; *Osawa et al., 2019*; *Valverde et al., 2019*) on ATG8 lipidation.

We further identified a distinct conformation of lipidated ATG8 proteins, in which their N-termini are inserted into membrane in *cis* (*Figures 2–4*). It should be noted that the *cis*-membrane association of ATG8 protein requires lipidation. The rescue assay using Hexa KO cells with non-tagged ATG8 WT or ATG8 membrane-expansion mutants showed that the *cis*-membrane association of ATG8 proteins are critical for autophagosome size regulation but not for cargo degradation (*Figures 6 and 7*). Based on this evidence, we propose that the N-terminal regions of ATG8 proteins play critical roles in autophagic membrane expansion by associating with and being inserted into the lipid bilayer (*Figure 8*).

It was previously proposed that phagophore expansion is mediated by membrane tethering and fusion driven by ATG8 proteins anchored on two distinct membranes ('in *trans*') (*Nakatogawa et al., 2007*; *Weidberg et al., 2011*; *Wu et al., 2015*). In this *trans* model, the N-terminus of lipidated Atg8 is expected to be solvent exposed. In support of this model, a recent study in yeast has resolved and modelled the structure of lipidated Atg8 on nanodiscs, revealing that two aromatic residues (F77/F79) of Atg8 are inserted into the membrane where the lipidated Atg8 resides, thereby facilitating the formation of tubulovesicular structures (*Maruyama et al., 2021*). In this study, the Atg8 N-terminus faces the cytoplasm, aligning with the '*trans*' model, in contrast to our model of *cis*-membrane association of ATG8 N-terminus (i.e., into the membrane where the lipidated ATG8 is located). These models can be reconciled if ATG8 adopts distinct conformational states on autophagic membranes. Indeed, it has been reported that the N-termini of ATG8 proteins adopt open and closed conformations in the non-lipidated state (*Coyle et al., 2002*; *Wu et al., 2015*).

Given that ATG8 proteins are widely distributed on autophagic membranes (*Sakai et al., 2020*) and present during the progression from early to late stages of autophagosome formation, lipidated ATG8 proteins may undergo substantial conformational changes depending on their localisation and the stages of phagophore growth. The convex and concave shapes of the phagophore membranes might be maintained by distinct conformational states of ATG8 proteins. Likewise, changes in membrane curvatures during autophagosome formation might have an impact on ATG8 conformation. Besides the membrane geometry and curvature, the membrane lipids themselves could be key players, particularly PE. The ATG8-dependent membrane fusion and hemifusion activities require a high PE concentration, which is unusual in cell membranes (*Nair et al., 2011*). Our data show that lipid-packing defects increased membrane association of the ATG8 N-termini (*Figure 3D*). Accordingly, the membrane environment may determine the ATG8 conformation. How the structural dynamics of lipidated ATG8 is mediated by membrane curvature and membrane environment during autophagosome formation will be interesting to probe in future research.

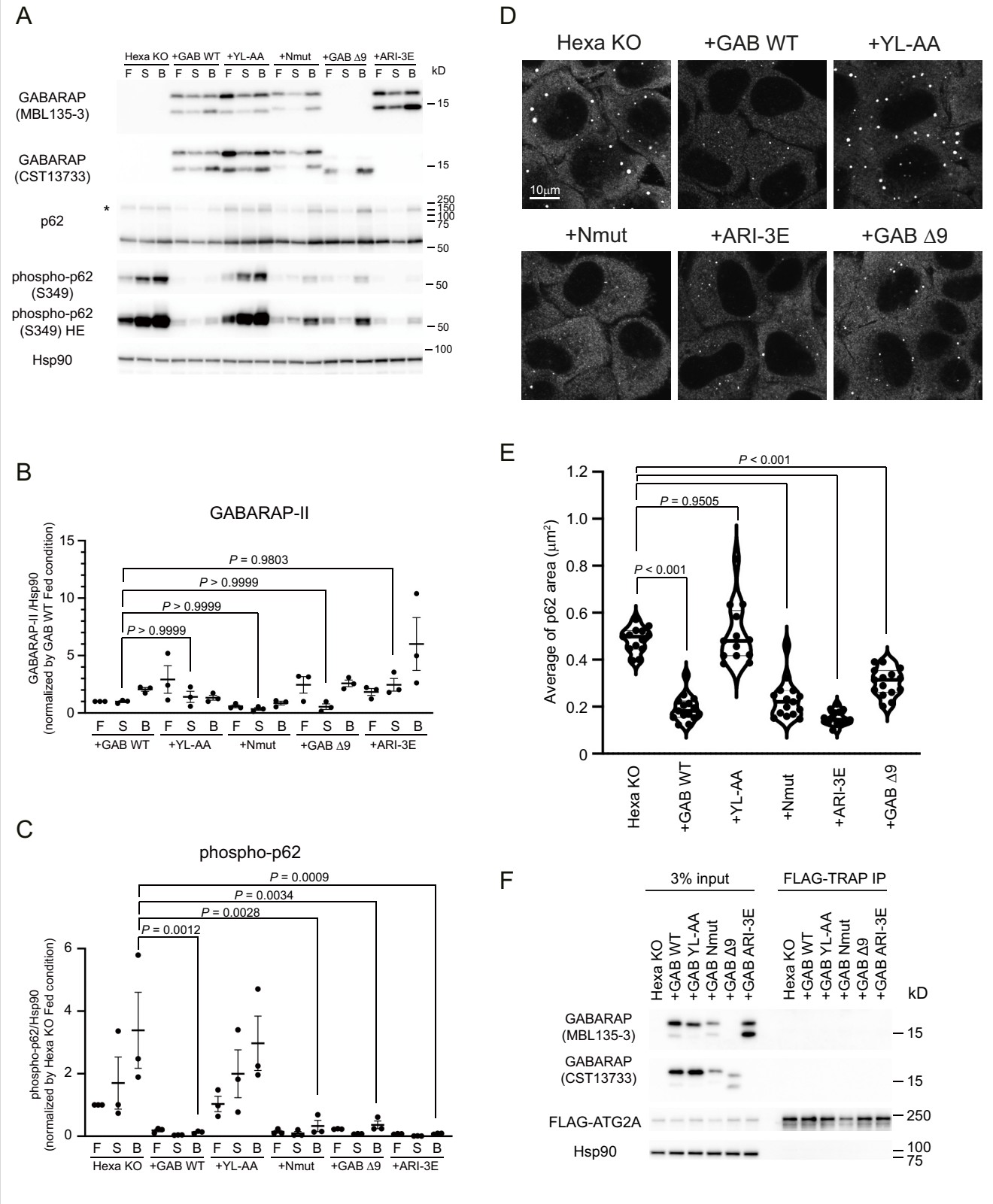

**Figure 6.** GABARAP N-terminus can be dispensable for the degradation of p62 body. (**A**) Hexa KO cells stably expressing non-tagged GABARAP WT, YL-AA, Nmut, Δ9, or ARI-3E were starved for 8 hr with (**B**) or without 100 nM Bafilomycin A$_1$ (S), or cultured in full media (F). Cell lysates were analysed by immunoblotting using the indicated antibodies. The asterisk indicates the position of high molecular weight forms of p62. (**B, C**) Band intensity quantification of GABARAP-II (**B**), and phosphorylated p62 (Ser349) (**C**). All data were normalised with those of HSP90. Data represent the mean ± SEM

*Figure 6 continued on next page*

*Figure 6 continued*

of three independent experiments. (**D**) The cells were starved for 2 hr before p62 (red) was visualised. Scale bar, 10 µm. (**E**) The violin plot of average p62 area. The thick and thin lines in the violin plot represent the medians and quartiles, respectively. n = 13 for Hexa KO and GAB YL-AA; n = 15 for GAB WT, Nmut, Δ9, and ARI-3E. (**F**) Co-immunoprecipitation experiments of FLAG-ATG2A with GABARAPs. Cell lysates were subjected to IP using anti-FLAG magnetic beads. The resulting precipitates were examined by immunoblot analysis with the indicated antibodies. Differences were statistically analysed by one-way ANOVA and Tukey multiple-comparison test. See also *Figure 6—figure supplement 1*.

The online version of this article includes the following source data and figure supplement(s) for figure 6:

**Source data 1.** Uncropped blot images of *Figure 6A and F*.

**Figure supplement 1.** Band intensity quantification of high molecular weight forms of p62.

How does membrane insertion of ATG8 N-terminus induce phagophore expansion? One possible explanation is that membrane area is increased by space occupation and membrane deformation (*Bangham, 1972*; *Maruyama et al., 2021*). As ATG8 proteins are abundant on autophagic membranes, insertion of their N-termini into the membrane could have a large impact on autophagosome size (*Figure 8*). Another scenario is that local membrane elasticity on each side of the phagophore membrane might be regulated by ATG8 N-terminus, which would help to facilitate membrane expansion (*Sakai et al., 2020*).

There is growing evidence that phagophore membrane growth is directed via the LIR binding of cargo receptors, such as p62 and NBR1, to lipidated ATG8 (*Kageyama et al., 2021*). This event, called wetting process, is crucial for efficient incorporation of specific cargos, like liquid-like droplets, into autophagosomes (*Agudo-Canalejo et al., 2021*). In this model, autophagy cargos serve as a template for phagophore membrane expansion. Previous reports suggest that the N-termini of ATG8 proteins are crucial for p62 recruitment in autophagy-lysosome degradation (*Shvets et al., 2008*; *Shvets et al., 2011*). Here, by rescuing HeLa Hexa KO cells with non-tagged GABARAPs, we showed that deletion of the N-terminus or *cis*-membrane association mutants of GABARAP was able to fully restore degradation of p62 bodies upon starvation, whereas the LIR docking mutant of GABARAP, which abolished the LIR-dependent p62 binding, was unable to rescue degradation of p62 bodies (*Figure 6D*). Therefore, inhibition of p62 degradation resulted from an abolished ATG8-cargo receptor interaction, whereas the clearance of p62 bodies did not requires GABARAP N-terminus and its *cis*-membrane association.

These results reinforce the previous observations (*Nguyen et al., 2016*) and are the first reported rescue experiments in HeLa Hexa KO using untagged ATG8s. Our results reveal a distinct function of ATG8 N-terminus solely in phagophore membrane expansion. In bulk non-selective autophagy, the phagophore randomly encapsulates cytosol, and it has been debated whether phagophore growth is or is not dependent on a cargo template. We show that rescue with the LIR-docking mutant of GABARAP (YL-AA) results in morphologically normal autophagosomes (*Figure 7*). In contrast, the Hexa KO cells expressing the GABARAP *cis*-membrane association mutants had smaller autophagosomes supporting the crucial role of *cis*-membrane association in expansion. These mutants also allowed us to distinguish the function of the N-terminus in *cis*-membrane association from its interaction with p62 and support the hypothesis that autophagosome formation can be independent of cargo. Our findings have broader implications on the molecular mechanism of autophagosome formation and membrane growth, explaining why phagophore expansion requires ATG8 lipidation.

## Materials and methods
### Plasmids

Descriptions of plasmids used in this study are given in *Supplementary file 1*. For in vitro lipidation assay, pAL-GST-LC3B and pAL-GST-GABARAP were truncated to expose the C-terminal glycine (LC3B G120 and GABARAP G116). GABARAP-His$_6$ construct was generated by introducing a His$_6$-tag after G116 in pAL-GST-GABARAP construct. LC3B-His$_6$ construct was generated by amplifying DNA sequence encoding LC3B (aa 1–120) and subcloning the sequence into pGEX-6P1 vector with a His$_6$-tag. All the point mutations in the recombinant LC3B and GABARAP were generated by Q5 Site-Directed Mutagenesis Kit (New England Biolabs). pGEX-6P1-GST-ATG3 was a gift from Dr. Alicia Alonso. cDNA encoding full-length human ATG7 (provided by Dr. Alicia Alonso, Instituto Biofisika, University of the

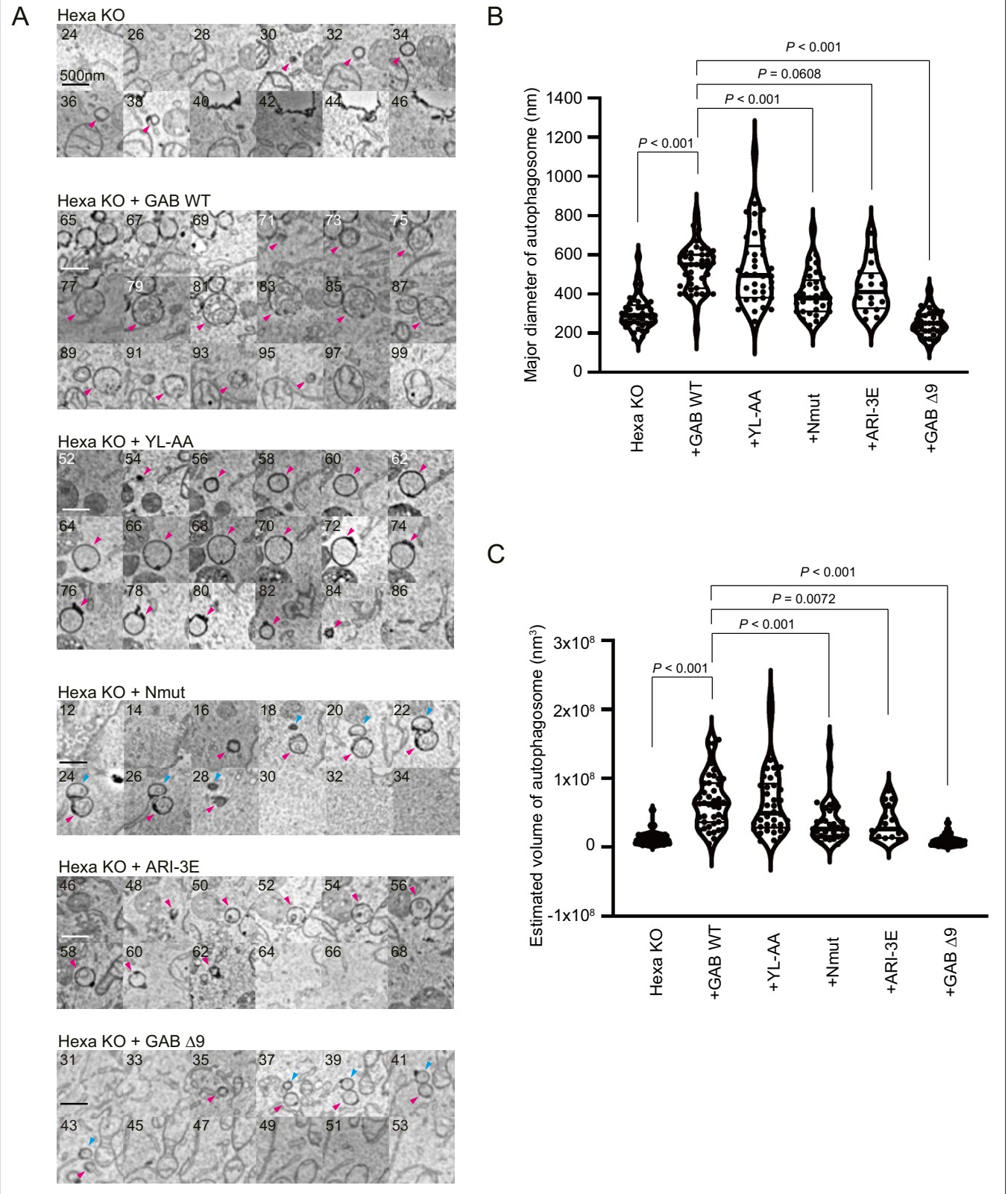

**Figure 7.** Autophagosome size is regulated by the N-terminus and *cis*-membrane association of GABARAP protein. Three-dimensional correlative light and electron microscopy (3D-CLEM) analysis. The starved cells were fixed, imaged by confocal microscopy, and subsequently relocated and imaged by scanning electron microscopy. (**A**) Consecutive 25 nm SEM slice images are shown. The number in each image indicates the slice number. The arrowheads show autophagosomal structures. Scale bars, 500 nm. (**B, C**) Size distributions of autophagosomes. Length, width, and height of each

*Figure 7 continued on next page*

Figure 7 continued

autophagosome are measured. The major diameter (**B**) and estimated volume (**C**) of each autophagosome are shown. The thick and thin lines in the violin plot represents the medians and quartiles, respectively. n = 42 for Hexa KO; n = 43 for GAB WT; n = 38 for YL-AA; n = 33 for Nmut; n = 16 for ARI-3E; n = 33 for Δ9. See also *Figure 7—figure supplement 1*.

The online version of this article includes the following figure supplement(s) for figure 7:

**Figure supplement 1.** Correlative light and electron microscopy (CLEM) images of STX17TM-positive autophagosomes.

Basque Country) was amplified by PCR and subcloned into pBacPAK-His$_3$-GST construct (provided by Dr. Svend Kjaer, The Francis Crick Institute) via In-Fusion cloning method (Takara Bio). StrepII$^{2x}$-ATG5, ATG7, ATG10, ATG12, and ATG16L1 were cloned into a single pFBDM plasmid.

For generation of stable retroviral transduced cell lines, cDNA encoding GABARAP (NM_007278) was amplified by PCR and subcloned into pMRX-IP backbone vector by Gibson Assembly (New England Biolabs, E2611L). All deletion and point mutation mutants of GABARAP were generated by a PCR-based method.

## Antibodies and reagents

Antibodies used for immunoblotting are listed as follows: rabbit polyclonal anti-ATG7 (Cell Signaling Technology, 2631), rabbit polyclonal anti-ATG3 (Sigma, A3231), rabbit polyclonal anti-LC3B (Abcam, ab48394), rabbit monoclonal anti-GABARAP (Cell Signaling Technology, 13733), mouse monoclonal anti-GABARAP (MBL, M135-3), mouse monoclonal anti-HSP90 (BD Transduction Laboratories, 610419), anti-FLAG M2 (Sigma, F1804), rabbit polyclonal anti-p62 (MBL, PM045, lot 022), rabbit polyclonal anti-Phospho-p62 Ser351 (MBL, PM074), and rabbit polyclonal anti-Actin (Abcam, ab8227). Secondary antibodies are HRP-conjugated anti-rabbit IgG (Jackson ImmunoResearch Laboratories, 111-035-144, or GE Healthcare, #NA934), and HRP-conjugated anti-mouse IgG (Jackson ImmunoResearch Laboratories, 315-035-003, or GE Healthcare, #NA931).

Antibodies used for immunofluorescence are listed as follows: rabbit polyclonal anti-p62 (MBL, PM045, lot 022) and Alexa Fluor 568-conjugated anti-rabbit IgG (Invitrogen, A-11036).

All lipids were purchased from Avanti: POPC (850457C), DOPC (850375C), POPE (850757C), DOPE(850725C), DGS-Ni-NTA (790404C), Rhod-PE (810150C), and Liver-PI (840042C). ATP solution (100 mM) was obtained from Thermo Fisher (R0441). IANBD was purchased from Invitrogen (D2004).

## Recombinant protein expression and purification

ATG3, LC3B, GABARAP, and their corresponding mutants were transformed in *Escherichia coli* BL21 (DE3) cells. Bacteria were grown in LB until $OD_{600} = 0.8$ and protein expression was induced with 0.5 mM IPTG for 16 hr at 18°C with all the constructs, except that in the case of GABARAP K2C ARI-3E, bacteria were grown in TB and protein expression was induced when $OD_{600} = 1.2$. These GST-tagged proteins were purified as described before with minor modifications (*Wirth et al., 2019*). Briefly, cells were harvested by centrifugation and resuspended in 50 mM Tris-HCl pH8.0, 500 mM NaCl, 0.5 mM TCEP, 0.4 mM AEBSF, and 15 µg/ml benzamidine. Cells were lysed by freeze–thaw followed by sonication. Lysates were then cleared by centrifugation at 25,000 × *g* for 30 min at 4°C. The GST-tagged proteins were absorbed with Glutathione-Sepharose 4B affinity matrix (GE Healthcare) for 1.5 hr and recovered by 3C protease cleavage at 4°C overnight in 50 mM Tris-HCl pH 8.0, 500 mM NaCl, and 0.5 mM TCEP. The proteins were further purified by size-exclusion chromatography using Superdex 200 16/60 column (GE Healthcare) equilibrated in buffer containing 25 mM Tris-HCl pH8.0, 150 mM NaCl, and 0.5 mM TCEP.

To purify human ATG7, pBacPAK-His$_3$-GST-ATG7 was transfected into insect cells Sf9 using the FlashBAC baculovirus expression system (Oxford Expression Technologies [OET]) and Fugene HD transfection reagent (Promega) according to the manufacturer's instructions. Virus at P1 was harvested after 5 d from transfection. 50 ml of Sf9 cells were infected with 1.5 ml of P1 virus to amplify the virus and harvest the stock virus P2. 200 ml of Sf9 cells (1.5–2 × 10$^6$ cells/ml) was infected with 1 ml of P2 virus and harvested after 60 hr. The cell pellets were frozen in liquid nitrogen and stored at –80°C until purification. When purifying ATG7, the cells were thawed on ice and resuspended in lysis buffer containing 50 mM Tris-HCl pH 8.0, 500 mM NaCl, 0.5 mM TCEP and EDTA-free cOmplete Protease Inhibitor cocktail (Roche). The suspension was sonicated 10 s, five times on ice, and then centrifuged

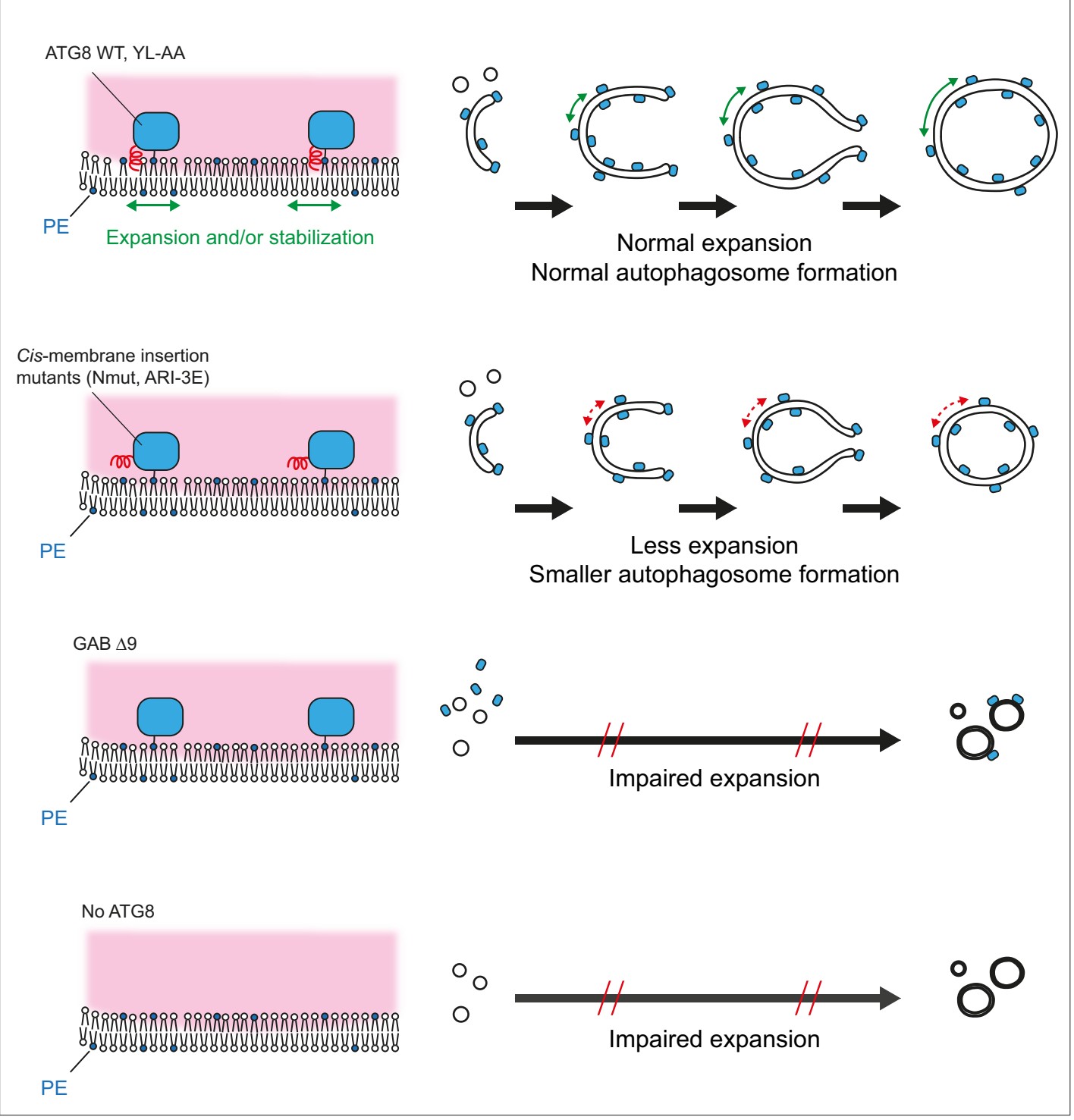

**Figure 8.** Model of autophagic membrane expansion mediated by lipidated mammalian ATG8 proteins. After ATG8 lipidation, their N-termini are inserted into autophagic membranes. The *cis*-membrane association of the N-termini promotes autophagic membrane expansion by inducing membrane area expansion or stabilising the membrane. Reduced N-terminal *cis*-membrane association results in less autophagic membrane expansion, and consequently smaller autophagosomes form. Deletion of ATG8 N-terminus or lack of ATG8 proteins impairs autophagosome expansion.

at 30,000 × *g* for 30 min at 4°C to remove cell debris. The remaining purification steps were performed as described above.

To purify human E3 complex, pFBDM-ATG7-ATG10-ATG12-StrepII[2x]-ATG5-ATG16L1 was transformed into DH10Multibac cells. The bacmid was isolated and Sf9 cells transfected. The resultant baculovirus was further amplified using standard procedures. The ATG12–ATG5-ATG16L1 complex was expressed in High Five insect cells using Sf-900 II SFM medium (Thermo). Cells were infected with a multiplicity of infection (MOI) greater than 2. Cells were harvested after 2.5 d and lysed using precooled lysis buffer containing 50 mM Tris HCl pH 8.3, 220 mM NaCl, 5% glycerol, 2 mM DTT, EDTA-free Protease Inhibitor tablets (Roche), 2 mM EDTA, 0.2 mM PMSF, 1 mM benzamidine, and Pierce universal nuclease. Cells were lysed by sonication and spun at 20,000 rpm for 1 hr using a JA-20 rotor. The supernatant was loaded onto a StrepTactin column (QIAGEN) pre-equilibrated with wash buffer composed of 50 mM Tris HCl pH 8.0, 220 mM NaCl, 5% glycerol, and 2 mM DTT. The StrepTactin column was washed with 20 column volumes (CV) of wash buffer before the E3 complex was eluted with 5 CV wash buffer containing 2.5 mM desthiobiotin. E3-containing fractions were pooled and loaded onto a ResQ anion exchange chromatography column (GE Healthcare). The E3 complex was eluted by applying a salt gradient from 50 to 700 mM NaCl (ResQ buffer base: 20 mM HEPES-NaOH pH 8.0, 5% glycerol, and 2 mM DTT). Protein containing fractions were pooled, concentrated, and loaded on a HiLoad 16/600 Superose 6 size-exclusion chromatography column pre-equilibrated in SEC buffer (20 mM HEPES NaOH pH 7.4, 180 mM NaCl, 5% glycerol, and 2 mM DTT). The E3 complex was concentrated using Amicon Ultra concentrators, aliquoted, and flash-frozen in liquid nitrogen.

## Liposome preparations

Lipids were mixed at the desired molar ratio in chloroform, dried under nitrogen gas, and further vacuumed for 2 hr to remove the remaining solvent. The lipid film was rehydrated and resuspended in the assay buffer containing 25 mM Tris-HCl pH 8.0, 150 mM NaCl, and 0.5 mM TCEP, with vortexing. Large unilameller vesicles (LUVs) were generated by five times freeze–thaw cycles in liquid nitrogen and water bath. The LUVs then were extruded 10 times through 0.2 µm membrane followed by at least 20 times through 0.1 µm membrane (Whatman) using a Mini-Extruder (Avanti Polar Lipid). The final concentration of liposome was 2 mM. The size of LUVs was checked by Zetasizer Nano ZS (Malvern Instruments). The LUVs had an average diameter of 100 nm. All the in vitro fluorescence measurements (the real-time lipidation assay, fluorescence-based liposome binding assay, and FRET assay) were performed in the same assay buffer used for liposome preparation.

## Protein NBD labelling

NBD is an environment-sensitive fluorescence probe with low molecular weight. It has been commonly used to monitor the environmental changes of specific residues of proteins (*Raghuraman et al., 2019*). LC3B and GABARAP proteins with a single cysteine mutation were labelled as described with some modifications (*Nishimura et al., 2019*). The purified proteins were transferred to labelling buffer Tris-HCl pH7.5 20 mM, NaCl 150 mM using a PD Minitrap G-10 column (Cytiva), immediately prior to labelling. The proteins were then labelled with 20-fold excess of IANBD-amide at room temperature in dark for 1 hr. The reaction was terminated with 4 mM cysteine and excess of IANBD was removed by a PD Minitrap G-10 column, preequilibrated with 20 mM Tris-HCl pH 7.5, 150 mM NaCl, and 0.5 mM TCEP. Glycerol was added to the NBD-labelled protein to a final concentration of 20% for storage at –80°C. The concentration of NBD-labelled proteins was determined by Bradford and checked by SDS-PAGE. The concentrations of the labelled proteins ranged between 30 µM to 80 µM.

## Real-time lipidation assay

All the real-time lipidation assays were carried out at 37°C with three independent repeats. Briefly, the reaction mix (80 µl) except the ATP/MgCl₂ was prepared with 0.2 µM ATG7, 0.2 µM ATG3, 1 µM NBD-labelled LC3B/GABARAP, 0.05 µM the E3 complex, and 1 mM liposomes. The reaction mix was then transferred into 10 mm pathlength quartz cuvette and the NBD fluorescence (ex/em 468 nm/535 nm) was measured immediately using FP-8300 spectrofluorometer (JASCO). The excitation bandwidth was fixed to 5 nm, and the emission bandwidth was fixed to 10 nm. The total measurement time was set to 20 min with two time intervals: in the first 80 s, NBD fluorescence was recorded every 20 s; after 80 s, the fluorescence was recorded every 10 s until the end. ATP/MgCl₂ (final conc. 1 mM) was

added into the reaction mix between 60 s and 80 s to initiate the lipidation reaction. For the control group containing all the proteins and liposomes except for the ATP/MgCl$_2$, the same amount of buffer was added instead. The fluorescence increase ($\Delta$Em535 nm) at each time point was calculated by subtracting the NBD signal recorded from the control group, normalised to the time point at 80 s.

For step-by-step assay (*Figure 1F*), there were some modifications with data collection intervals. The total measurement time was set to 1320 s, and the NBD fluorescence was recorded every 20 s. There were four reaction groups: (1) hATG8$^{NBD}$, (2) hATG8$^{NBD}$ + liposomes, (3) hATG8$^{NBD}$ + ATP/MgCl$_2$, and (4) hATG8$^{NBD}$ + liposomes + ATP/MgCl$_2$. ATG7 (final conc. 0.2 μM) was added between 60 s and 80 s. ATG3 (final conc. 0.2 μM) was added between 480 s and 500 s. The E3 complex (final conc. 0.05 μM) was added between 900 s and 920 s. There were two control groups: (1) hATG8$^{NBD}$ and (2) hATG8$^{NBD}$ + liposomes. Instead of adding ATG7, ATG3, and the E3 complex, the same amount of buffer was added in between each corresponding time interval. Reactions (1) and (3) were calibrated to the control group (1), subtracting the background signal of hATG8$^{NBD}$. Reactions (2) and (4) were calibrated to the control group (2), subtracting the background signal of hATG8$^{NBD}$ and liposomes. We determined the fluorescence increase ($\Delta$Em535 nm) at each time point normalised to the fluorescence recorded at the time point after the addition of ATG7 (80 s), ATG3 (500 s), and the E3 complex (920 s).

## In vitro ATG8 lipidation assay

1 μM ATG7, 1 μM ATG3, 0.25 μM ATG12-ATG5-ATG16L1, 5 μM LC3B or GABARAP WT and their N-terminal deletion mutants LC3B ΔN11 and GABARAP ΔN9 and 1 mM liposomes were mixed in the assay buffer in a total volume of 10 μl. Lipidation reactions were initiated by adding 5 mM MgATP and then incubated for the indicated time (0 min, 10 min, 20 min, 60 min, and 90 min) at 37°C.

## Fluorescence-based liposome binding assay and FRET assay

The spectra of fluorescence-based liposome binding assay and FRET assay were recorded at 37°C. For fluorescence-based liposome binding assay, NBD-labelled LC3B-His$_6$/GABARAP-His$_6$ (1 μM) were mixed with liposomes (final conc.1mM) in a total volume of 80 μl and immediately measured using FP-8300 spectrofluorometer. The emission spectra were recorded from 500 nm to 650 nm by exciting NBD at 468 nm. The excitation and emission bandwidths were set to 5 nm. For FRET assays, the settings of fluorescence measurements were kept the same. The NBD-labelled LC3B-His$_6$/GABAR-AP-His$_6$ (1 μM) were mixed with blank liposomes or rhodamine liposomes (final conc. 1 mM) in the presence or absence of imidazole (final conc. 100 mM). The NBD fluorescence was recorded immediately. The spectra of buffer and all liposomes solutions were recorded as background control. The spectra of NBD-labelled proteins were corrected by subtracting the buffer spectra, while the reactions containing liposomes were corrected by subtracting the corresponding liposome spectra.

## Liposome tethering assay

Various concentrations (1–10 μM) of GABARAP-His$_6$ WT, ΔN9, and Nmut proteins were mixed with 1 mM liposomes in a total volume of 80 μl. The liposome turbidity was measured immediately using Shimadzu UV-2550 spectrophotometer, following the absorbance of liposomes at 405 nm and at 37°C for 10 min. To obtain the liposome tethering activity resulted from GABARAP-His$_6$ proteins, $\Delta$OD405 was calculated by subtracting the absorbance of liposomes from that of the experimental groups mixing liposomes and GABARAP-His$_6$ proteins.

## Molecular dynamic simulations

The protein structure of LC3B (3VTU) and GABARAP (1GNU) protein were used and residues after Gly-120 and Gly-116 were deleted (*Jatana et al., 2020*). A covalent bond between C-terminal glycine and phosphatidylethanolamine (PE) was introduced where the lipid anchor parameters were taken from already existing 1-palmitoyl-2-oleoly-*sn*-phosphoethanolamine (POPE) of charmm36 ff. The lipidated LC3B/GABARAP proteins were placed in a 400-lipid POPC bilayer membrane which was prepared using charmm-GUI web server. To obtain statistically accurate docking of LC3 on membrane, we undertook multiple conformations with varied lipid-contacting orientations (*Figure 2*). After an initial refinement of 100 ns, all conformations converged to a single orientation with N-terminal moving closer to lipids. Hence, the conformation with membrane facing N-terminal was taken for further

simulations. In total, three simulations each were started for LC3 and GABARAP and cumulative simulation length was 6.6 μs was obtained.

The lipidated structures of LC3B/GABARAP with POPC membrane were placed in a rectangular box large enough to accommodate protein and membrane. Water molecules were added with TIP3P representation and Na$^+$ ions were added to neutralise the systems (*Mark and Nilsson, 2001*). MD simulation was performed using GROMACS version 2018.3 by utilising charmm36 all-atom FF (*Abraham et al., 2015*). Periodic boundary conditions were used, and 1.2 nm was set as real space cut-off distance. Particle Mesh Ewald (PME) summation using the grid spacing of 0.16 nm was used in combination with a fourth-order cubic interpolation to deduce the forces and potential in-between grid points (*Darden, 1993*). The Van der Waals cut-off was set to 1.2 nm. Energy minimisation was performed on the initial systems using the steepest descent method and the temperature and pressure were maintained at 310 K and 1 bar using Nose–Hoover thermostat and Parrinello–Rahman barostat, respectively (*Bussi et al., 2007*; *Parrinello and Rahman, 1981*). A time step of 2 fs was used for numerical integration of the equation of motion. The coordinates were saved at every 20 ps. Three replicas for each lipidated LC3B and GABARAP systems were simulated for 1 μs. All the molecular images were rendered using UCSF Chimera and VMD (*Humphrey et al., 1996*). The graphs and plots were generated using MATLAB and Python libraries.

## Analysis of trajectories

The periodic boundary conditions were removed before performing analysis on the trajectories. The distance was calculated between the centre of mass of residues, and the POPC membrane was calculated using the gmx distance module which calculates the distance between two positions as a function of time. The probability of contact formation was calculated by defining a contact when the distance is between residues and membrane was <3.5 nm. Furthermore, the contacts were categorised into three parts depending upon the residues' side chain orientation on the membrane: inserted at ≤2.3 nm, at membrane surface at ≤2.8 nm, and in proximity at <3.5 nm.

## Cell lines and culture conditions

Authenticated human embryonic kidney (HEK) 293T cells were used in this study. Hexa KO cell line was generated previously (*Nguyen et al., 2016*). Cells were maintained in Dulbecco's Modified Eagle Medium (DMEM) (Wako, 043-30085) supplemented with 10% foetal bovine serum (FBS) (Sigma-Aldrich, 173012) in a 5% CO$_2$ incubator at 37°C. Hexa KO cells stably expressing non-Tagged GABARAPs were generated as follows: HEK293T cells were transfected using Lipofectamine 2000 reagent (Thermo Fisher Scientific, 11668019) with pMRX-IP-based retroviral plasmid, pCG-VSV-G and pCG-gag-pol, following which the medium was replaced with fresh medium. After 3 d, the culture medium was collected and filtered with a 0.45 μm filter unit (Millipore, SLHVR33RB). Hexa KO cells were treated with the retrovirus containing medium and 8 μg/ml polybrene (Sigma-Aldrich, H9268). After two passages, cells were incubated with full medium containing 50 nM SNAP-TMR for 20 min and then sorted by flow cytometry (Sony, SH800) to select non-Tagged GABARAPs-expressing cells. In the case of GFP-Stx17TM, GFP-positive cells were selected after retrovirus infection.

## Immunoblotting

Cells were cultured under DMEM supplemented with FBS or DMEM without amino acids (Wako, 048-33575) in the absence or presence of 100 nM bafilomycin A$_1$ (Invitrogen, B1793) for 8 hr, collected in ice-cold PBS by scraping and then precipitated by centrifuged at 1000 × *g* for 3 min. The precipitated cells were suspended in 100 μl lysis buffer (25 mM HEPES-NaOH, pH 7.5, 150 mM NaCl, 2 mM MgCl$_2$, 0.2% *n*-dodecyl-b-D-maltoside [nacalai, 14239-54] and protease inhibitor cocktail [nacalai, 03969-34]) and incubated on ice for 20 min. 90 μl of cell lysates were mixed with 10 μl of lysis buffer containing 0.1 μl benzonase (Merck Millipore, 70664) and further incubated on ice for 15 min. The remaining cell lysates were centrifuged at 17,700 × *g* for 15 min, and the supernatant was used to measure protein concentration by NanoDrop One spectrophotometer (Thermo Fisher Scientific). 100 μl of cell lysates were mixed with SDS-PAGE sample buffer and heated at 95°C for 5 min. Samples were subsequently separated by SDS-PAGE and transferred to Immobilon-P PVDF membranes (Merck Millipore, IPVH00010) with Trans-Blot Turbo Transfer System (Bio-Rad). After incubation with the indicated antibodies, the signals from incubation with SuperSignal West Pico PLUS Chemiluminescent Substrate

(Thermo Fisher Scientific, 34580) was detected with Fusion Solo S (VILBER). Band intensities were quantified with Fiji.

For in vitro real-time lipidation reaction, 20 µl reaction mix was taken immediately after the reaction, mixed with 5 µl SDS-PAGE sample buffer, and heated at 95°C for 5 min. 10 µl reaction mix was resolved on NuPAGE Bis-Tris 4–12% gels (Life Technologies) followed by immunoblotting.

## Immunoprecipitation

Cells were lysed in ice-cold lysis buffer containing 50 mM Tris-HCl 7.4, 150 mM NaCl, 1 mM EDTA, 1 mM DTT, 0.5% w/v Triton X-100, and protease inhibitor cocktail (nacalai). Cell lysate was centrifuged at 17,700 × $g$ for 15 min. FLAG-ATG2A proteins were immunoprecipitated using anti-FLAG M2 magnetic beads (Sigma, M8823), incubating for 2 hr at 4°C. After washing beads five times with lysis buffer, beads were mixed with SDS-PAGE sample buffer and boiled at 95°C for 5 min. The proteins of interest were resolved on SDS-PAGE gel followed by western blotting.

## Fluorescence microscopy

Cells grown on coverslips were fixed with 4% paraformaldehyde in PBS for 15 min, permeabilised with 50 µg/ml digitonin (D141; Sigma-Aldrich) in PBS for 5 min, blocked with 3% BSA in PBS for 30 min, and then incubated with anti-p62 antibody for 1 hr. After washing five times with PBS, cells were incubated with Alexa Fluor 568 conjugated goat anti-rabbit IgG secondary antibody for 1 hr. These specimens were observed using a confocal FV3000 confocal laser microscope system (Olympus). For the final output, images were processed using Adobe Photoshop 2021 v22.3.1 software (Adobe). Average of p62 were measured using the open-source software Fiji.

## Correlative light and electron microscopy

Correlative light and electron microscopic (CLEM) analysis was performed as previously described (*Maeda et al., 2020*; *Morishita et al., 2021*). In brief, Hexa KO stably expressing GFP-STX17TM, were grown for 48 hr in gridded coverslip-bottom dishes (custom made by attaching the coverslips upside down based on a dish IWAKI 3922-035). The cells were fixed and observed by FV3000 confocal laser microscope system (Olympus) equipped with a ×60 oil-immersion objective lens (NA1.4, PLAPON60XOSC2; Olympus). After fluorescent-image acquisition, the cells were embedded in EPON812 (TAAB) for electron microscopic observation as previously described (*McArthur et al., 2018*; *Morishita et al., 2021*). After embedding and removing the coverslip, the resin block was trimmed to retain the area imaged by confocal laser scanning microscope (about 150 × 150 µm). Then the block was sectioned using ultramicrotome (EM UC7, Leica) equipped with a diamond knife (Ultra JUMBO, 45°, DiATOME) to cut 25 nm serial sections using an active vibration isolation table and an eliminator of the static electricity to make long serial sections. Then the sections were collected on a cleaned silicon wafer strip held by a micromanipulator (Märzhäuser Wetzlar). Sections were directly imaged (without staining) using a scanning electron microscope (JSM7900F, JEOL) supported by a software (Array Tomography Supporter, System in Frontier). Images were stacked in order using the software (Stacker NEO, System in Frontier). Correlation of light and electron microscopic images was performed by Fiji software or Adobe Photoshop. Before quantification of autophagosome sizes, the samples were shuffled for a randomised double-blind analysis. The diameters of autophagosomal structures in the maximum cross section were measured by Fiji software. The total number of slices showing autophagosomes was used to measure their heights. Autophagosome volume was calculated using the formula for volume of ellipsoid sphere: V = length × width × height × π/6.

## Data analysis

Differences were statistically analysed by one-way ANOVA and Tukey multiple-comparison test. Statistical analysis was carried out using GraphPad Prism 9 (GraphPad Software).

## Acknowledgements

We thank Noboru Mizushima (NM), the director of Exploratory Research for Advanced Technology (ERATO) Mizushima Intracellular Degradation project, for helpful discussion; Yoko Ishida for cutting ultrathin sections and providing electron microscopy pictures; Ikuko Koyama-Honda, Satoru Takahashi, and Keiko Igarashi for technical assistance with the 3D-CLEM experiments; Shoji Yamaoka for pMRXIP;

and Teruhito Yasui for pCG-VSV-G and pCG-gag-pol. We thank Michael Lazarou for providing Hexa KO cell line, Alicia Alonso for human ATG7 and ATG3 plasmids, Svend Kjaer for pBacPAK-His$_3$-GST plasmid and the technical assistance with insect cell culture, Simone Kunzelmann for the technical assistance and helpful suggestions on fluorescence spectroscopy, and Stefano De Tito for critical comments and suggestions. We thank the support from CSIR-IGIB for infrastructure and CSIR-4PI for supercomputing facilities. For the purpose of Open Access, the author has applied a CC BY public copyright licence to any Author Accepted Manuscript version arising from this submission.

## Additional information

### Funding

| Funder | Grant reference number | Author |
| --- | --- | --- |
| European Research Council | FP7/2007-2013 788708 | Wenxin Zhang<br>Sharon A Tooze |
| Wellcome Trust | CC2134 | Wenxin Zhang<br>Harold BJ Jefferies<br>Sharon A Tooze |
| Wellcome Trust | CC2064 | Colin Davis<br>Anne Schreiber |
| Cancer Research UK | CC2134 | Wenxin Zhang<br>Harold BJ Jefferies<br>Sharon A Tooze |
| Cancer Research UK | CC2064 | Colin Davis<br>Anne Schreiber |
| Medical Research Council | CC2134 | Wenxin Zhang<br>Harold BJ Jefferies<br>Sharon A Tooze |
| Medical Research Council | CC2064 | Colin Davis<br>Anne Schreiber |
| Japan Science and Technology Agency | PRESTO JPMJPR20EC | Taki Nishimura |
| Japan Society for the Promotion of Science | Grant-in-Aid for Transformative Research Areas (B) 21H05146 | Taki Nishimura |
| Japan Science and Technology Agency | ERATO JPMJER1702 | Chieko Saito |
| Council of Scientific and Industrial Research, India | OLP1163 | Deepanshi Gahlot<br>Lipi Thukral |

The funders had no role in study design, data collection and interpretation, or the decision to submit the work for publication. For the purpose of Open Access, the authors have applied a CC BY public copyright license to any Author Accepted Manuscript version arising from this submission.

### Author contributions

Wenxin Zhang, Conceptualization, Formal analysis, Validation, Investigation, Methodology, Writing - original draft, Project administration, Writing - review and editing; Taki Nishimura, Conceptualization, Formal analysis, Funding acquisition, Validation, Investigation, Methodology, Writing - original draft, Project administration, Writing - review and editing; Deepanshi Gahlot, Chieko Saito, Conceptualization, Formal analysis, Validation, Investigation, Visualization, Methodology, Writing - original draft, Writing - review and editing; Colin Davis, Validation, Visualization; Harold BJ Jefferies, Validation, Investigation, Writing - review and editing; Anne Schreiber, Validation, Investigation, Visualization, Writing - review and editing; Lipi Thukral, Conceptualization, Supervision, Funding acquisition, Validation, Investigation, Visualization, Writing - original draft, Project administration, Writing - review and editing; Sharon A Tooze, Conceptualization, Resources, Supervision, Funding

acquisition, Validation, Investigation, Writing - original draft, Project administration, Writing - review and editing

## Author ORCIDs

Wenxin Zhang ⓘ http://orcid.org/0000-0002-7657-4495
Taki Nishimura ⓘ http://orcid.org/0000-0003-4019-5984
Deepanshi Gahlot ⓘ http://orcid.org/0000-0002-2681-8818
Lipi Thukral ⓘ http://orcid.org/0000-0002-1961-039X
Sharon A Tooze ⓘ http://orcid.org/0000-0002-2182-3116

## Decision letter and Author response

Decision letter https://doi.org/10.7554/eLife.89185.sa1
Author response https://doi.org/10.7554/eLife.89185.sa2

---

## Additional files

### Supplementary files
• MDAR checklist
• Supplementary file 1. Plasmids used in this study.

### Data availability
All data generated or analysed during this study are included in the manuscript and supporting file. Source data files have been provided for Figures 1–6.

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
