## [Editor Report]

In this study, the exciting possibility that Atg8s act on the membrane in "cis" is explored. While the correlation between in vitro data, MD simulations, and cell biology experiments could be further strengthened, the study presents a compelling case for giving serious consideration to the "cis" model.

---

## [Decision Letter]

**Decision letter after peer review:**

[Editors’ note: the authors submitted for reconsideration following the decision after peer review. What follows is the decision letter after the first round of review.]

Thank you for submitting the paper "Cis-membrane association of human ATG8 proteins N-terminus mediates autophagy" for consideration by *eLife*. Your article has been reviewed by 3 peer reviewers, and the evaluation has been overseen by a Reviewing Editor and a Senior Editor. The reviewers have opted to remain anonymous.

We are sorry to say that, after consultation with the reviewers, we have decided that this work will not be considered further for publication by *eLife*, at least in the current form. I feel that the points raised by the reviewers are more substantial than can be addressed in a typical revision period. If you wish to expedite the publication of the current data, it may be best to pursue publication in another journal.

Given the interest in the topic, we would be open to resubmission to *eLife* of a significantly revised and extended manuscript that fully addresses the reviewers' concerns and is subject to further peer review. Please note that priority and novelty would be reassessed at resubmission.

*Reviewer #1 (Recommendations for the authors):*

The manuscript by Zhang and Nishimura et al. addresses the roles of ATG8 family proteins in autophagosome biogenesis. Autophagosomes are formed de novo upon induction of autophagy and sequester cytoplasmic material, such as the p62 protein, within double membrane vesicles for degradation within lysosomes. ATG8 proteins including LC3B and GABARAP are covalently attached to lipids residing in the autophagosomal membrane in a ubiquitin-like conjugation reaction. They are required for efficient autophagosome formation and cargo degradation. Various functions such as direct membrane remodeling, tethering, and fusion activities as well as interactions with cargo receptors (for example p62) and components of the autophagy machinery have been attributed to ATG8s. However, their functions are not fully understood.

Here the authors employ a novel in vitro lipidation assay in combination with molecular dynamics simulations and cell biological approaches to obtain further insights into the mechanisms of action of ATG8 proteins during autophagosome biogenesis. Their results suggest that the N-termini in addition to other regions of these proteins insert cis into the same membrane to which they are lipidated. Correlating their in vitro studies and MD simulations with cell biology experiments, the authors suggest that the insertion of the ATG8 N-termini in cis is important for membrane expansion and efficient capture of p62 bodies. Overall, this is a comprehensive study providing novel insights into the important question of how the ATG8 proteins contribute to autophagosome biogenesis. In addition, the real-time in vitro assay to follow ATG8 membrane insertion has the potential for wider use in the field.

Concerns

1. My main concern regarding the manuscript is the correlation between the in vitro data, the MD simulations, and the cell biology experiments. It is convincingly shown that the N-termini of LC3B and GABARAP insert into the membrane in cis (not excluding any trans interaction). In order to test if this membrane insertion is important in cells, the N-terminus is deleted (delta9/11) showing that the N-termini are indeed important for autophagosome biogenesis. However, it is unclear if these deletion mutants are lipidated with the same efficiency. This could be tested in their in vitro system. Also, these proteins have been shown to interact with various cargo receptors such as p62 and components of the autophagy machinery. It is already shown in this manuscript that the N-terminal deletion mutants display weakened interactions with p62 (Figure 5H). The same may be true for various other interaction partners. Therefore, the observed defects may be due to reduced protein-protein interactions or even to reduced membrane tethering/fusion activities in which the N-termini has also been implicated. The authors should therefore attempt to identify point mutation in the N-termini of the ATG8 proteins, which lose cis membrane interactions and test those in cells. This would give greater confidence that loss of cis membrane interaction is indeed the cause of the autophagy defects observed for the deletion mutants.

2. Regarding the lipid composition used for the in vitro assays, the authors should consider including some PS, PI, or PI3P. These are negatively charged lipids and are present in the phagophore. They may therefore influence the orientation of the ATG8 proteins on the membrane, in particular with respect to RRR/RKR regions.

3. The authors should discuss/mention that the MD simulations do of course favor cis vs. trans interactions with the membrane because as far as I understand no transmembrane (with or without ATG8 proteins) was included. Thus, it is possible that trans interactions are preferred over cis interactions once it becomes available.

*Reviewer #2 (Recommendations for the authors):*

Atg8 proteins play a dual role in autophagosome formation. On one hand, Atg8s interact with cargo receptors to facilitate cargo recruitment into the inner space of the forming autophagosome. And on the other hand, these proteins were also implicated in the membrane expansion of the phagophore. The exact mechanism responsible for the later activity is still largely unclear. Using liposome assays it has been demonstrated by different research groups, that upon conjugation to PE on the membrane, Atg8s mediate membrane tethering and fusion. However, this view has been challenged as membrane fusion of lipidated Atg8 may require a large amount of PE or a large amount of Atg8 per membrane.

In the present study, Zhang et al. utilized a liposome-based assay to determine the mechanism by which GABARAP and LC3B, two members of the Atg8 family, perturbate the membrane upon conjugation to PE. The authors used NBD-labeled Atg8s at different sites to follow changes in NBD fluorescence upon conjugation to the membrane. Conjugation in this system was mediated either enzymatically by the conjugation machinery in the presence of ATP, or chemically, using His6 C-terminus Atg8s and Ni-NTA lipids. Using these systems, the authors found that NBD placed at the N-terminus of the protein indeed changes its fluorescence upon conjugation. The authors also utilized yet another cell-free assay that is based on FRET to measure the interaction between conjugated GABARAP or LC3B with the lipids. This system too indicated a direct interaction between the proteins' N-terminus and the membranes. Finally, the authors utilized the ability of their different GABARAP/LC3B mutants to overcome autophagy inhibition observed upon deletion of all six Atg8 family members. Interestingly, both GABARAP and LC3B recovered the block in autophagy measured by following lysosomal degradation of either p62 or the transfected Atg8 protein. This finding is somewhat surprising as Nguyen et al. JCB 2016 originally indicated that GABARAP proteins but not LC3 are essential for autophagosome formation and targeting of lysosome. More importantly, the authors show that GABARAP mutants lacking the N-terminus 9 amino acids or GABARAP mutated in loop 3 (ARI-3E), maintain the ability to target the mutant Atg8 protein to the lysosome while losing the ability to efficiently engulf p62. A CLEM analysis of the membrane intermediates accumulated by these mutants indicated a defect in their ability to elongate/expand. Based on these findings the authors propose that the "cis" insertion of Atg8 N-terminus to the bilayer is needed for such membrane expansion.

Overall, this is a clearly written manuscript and the data obtained from the cell-free systems are straightforward and convincing. The correlation between the cell-free findings and the experiments in tissue culture cells is reasonable, however, they serve as a rather weak support for the suggested model. A more direct in vivo analysis is needed.

The main message of this study is the exciting possibility that Atg8s are acting on the membrane in "cis". This is based almost solely, on molecular dynamics simulation. No attempt was made to utilize this method to determine the ability of the different mutants (particularly ARI-3E) used in this study to undergo such conformational changes. Moreover, it is not clear whether the in silico approach took into account the close proximity of another lipid bilayer in "trans". Along these lines, the authors need to directly test whether lipidation of their liposome mediates membrane tethering and fusion, which may imply such "trans" activity.

It should be noted that the idea that Atg8 can perturbate the membrane has been recently proposed by Maruyama et al. as indicated by the authors. However, while Maruyama et al. suggest that residues F77 and F79 on the yeast Atg8 protein are inserted into the lipid bilayer (based on NMR studies and GUVs) the current study indicates that the N-terminus is responsible for such perturbation. In both cases, the main conclusions are based on data obtained from cell-free systems, which may turn misleading.

The CLEM analysis shown in figure 6 is very impressive, showing that indeed GABARAP is needed for autophagosome expansion. While an important finding, this by itself is not completely new as these proteins were previously implicated in the growth of the autophagic membrane. As indicated above, the conclusion that this is mediated by "cis" acting Atg8 is not demonstrated by the current data. A more rigorous experimental approach aiming to address this directly is needed to solidify this model.

The authors' proposal that insertion of the GABARAP/LC3B N-terminus to the membrane will by itself lead to membrane expansion assumes a large ratio of Atg8 to lipid on autophagosomes. This has never been shown and the current study does not address this as well. Alternatively, an exciting possibility may be a link between these proteins on the autophagosomal membrane and Atg2 which was recently implicated in lipid transfer from the ER to the growing autophagosome. The authors are encouraged to test this directly.

As for the data shown in figure 5, changes in the p62 level may also result from changes in transcription. The authors should test this directly by RT-PCR or any other method. The IF analysis shown in panel F is missing the Baf. A control. Labeling the intermediates with additional markers such as WIPI, Atg2, etc. could help clarify their nature.

*Reviewer #3 (Recommendations for the authors):*

The authors examined an important topic, the membrane interactions of the mammalian ATG8 family proteins LC3B and GABARAP, using a combination of molecular dynamics simulations and site-specific NDB labeling, and they find that the N-terminal region of ATG8s binds tightly to membranes. This paper follows the Maruyama et al. study of yeast Atg8 using NMR that came to a different conclusion, that a patch near F77/79 (F80 in human LC3B), binds membranes. The present study concludes that F80 is far from the membrane. The discrepancy may be related to the authors' use of an uncharged lipid mixture in the present study, compared to the more physiological mixture containing 10% PI used by Maruyama. Further, Maruyama used Atg8 covalently conjugated by the physiological conjugation system, while the present study uses a hexahistidine tag to non-covalently tether LC3B and GABARAP to Ni-NTA-containing membranes. Apart from these major limitations in study design, the manuscript generally uses state-of-the-art experimental approaches, and a lot of work clearly went into it, thus it is quite unfortunate that non-optimal choices made in the initial design of the study have rendered the conclusions non-physiologically relevant, or at least, unconvincing.

1. The use of a charge-neutral liposome mixture in both experiments and simulations does not model the phagophore in a realistic way and likely imparts a strong bias to the results. This is the most serious flaw in the study and very likely accounts for the discrepancy with Maruyama. The simulations and NBD interactions should be redone with 10% PI (or some similar charged composition based on knowledge of the mammalian phagophore lipid composition).

2. The use of the C-terminal His6 tag construct in the experimental setup was an ill-advised choice in the experimental design, and may substantially perturb the results. The authors should perform the correct conjugation experiment with LC3B(G120) and GABARAP (G116) and redo all of the NDB probe experiments in this condition. The physical chemistry of the His tag linkage is very different from that of the normal amide linkage of the C-terminal Gly to the PE headgroup. The introduction of a nickel ion potentially confounds the measurement of the membrane associations. In this respect, the experiments do not match the simulations, so direct comparisons cannot be made.

3. The authors advance the idea that the membrane association of an amphipathic helix of ATG8's N-terminus is required for autophagosomal expansion in the case of P62 aggregates. The basic residues of the first α helix of LC3B interact with the acidic residues of p62's LIR domain. The authors' data shows that pull down with p62 is weakened by the loss of the N-terminal helix. A necessary control experiment, in this case, is for a measurement of the drop in affinity of p62 for LC3B and GABARAP, as the reduction of association between ATG8 and the cargo receptors could present an alternative explanation for the reduction in autophagosomal expansion efficiency for the p62 clearance case.

---

## [Author Response]

[Editors’ note: the authors resubmitted a revised version of the paper for consideration. What follows is the authors’ response to the first round of review.]

Given the interest in the topic, we would be open to resubmission to eLife of a significantly revised and extended manuscript that fully addresses the reviewers' concerns and is subject to further peer review. Please note that priority and novelty would be reassessed at resubmission.Reviewer #1 (Recommendations for the authors):The manuscript by Zhang and Nishimura et al. addresses the roles of ATG8 family proteins in autophagosome biogenesis. Autophagosomes are formed de novo upon induction of autophagy and sequester cytoplasmic material, such as the p62 protein, within double membrane vesicles for degradation within lysosomes. ATG8 proteins including LC3B and GABARAP are covalently attached to lipids residing in the autophagosomal membrane in a ubiquitin-like conjugation reaction. They are required for efficient autophagosome formation and cargo degradation. Various functions such as direct membrane remodeling, tethering, and fusion activities as well as interactions with cargo receptors (for example p62) and components of the autophagy machinery have been attributed to ATG8s. However, their functions are not fully understood.Here the authors employ a novel in vitro lipidation assay in combination with molecular dynamics simulations and cell biological approaches to obtain further insights into the mechanisms of action of ATG8 proteins during autophagosome biogenesis. Their results suggest that the N-termini in addition to other regions of these proteins insert cis into the same membrane to which they are lipidated. Correlating their in vitro studies and MD simulations with cell biology experiments, the authors suggest that the insertion of the ATG8 N-termini in cis is important for membrane expansion and efficient capture of p62 bodies. Overall, this is a comprehensive study providing novel insights into the important question of how the ATG8 proteins contribute to autophagosome biogenesis. In addition, the real-time in vitro assay to follow ATG8 membrane insertion has the potential for wider use in the field.

We thank the reviewer for their supportive comments.

Concerns1. My main concern regarding the manuscript is the correlation between the in vitro data, the MD simulations, and the cell biology experiments. It is convincingly shown that the N-termini of LC3B and GABARAP insert into the membrane in cis (not excluding any trans interaction). In order to test if this membrane insertion is important in cells, the N-terminus is deleted (delta9/11) showing that the N-termini are indeed important for autophagosome biogenesis. However, it is unclear if these deletion mutants are lipidated with the same efficiency. This could be tested in their in vitro system.

We thank the reviewer for the suggestions. We checked the lipidation efficiency of the N-terminal deleted LC3B (ΔN11) and GABARAP (ΔN9) with in vitro lipidation assays and included the results in the revised manuscript on page 5, 2^nd^ paragraph. As shown in the manuscript (new data in Figure 1—figure supplement 2), compared to the wild-type proteins, LC3B ΔN11 had a reduced lipidation efficiency, while GABARAP ΔN9 had a similar lipidation efficiency compared to wild-type GABARAP. Since the lipidation efficiency of LC3B/GABARAP is more susceptible to the PE concentration (Nath et al., Nat Cell Biol., 2010), we used liposomes containing high percentage of PE (50%). As suggested by the Reviewer’s comment 2, we also tested a lipid composition which mimics the ER lipid composition (“ERlike” in our revised context).

Also, these proteins have been shown to interact with various cargo receptors such as p62 and components of the autophagy machinery. It is already shown in this manuscript that the N-terminal deletion mutants display weakened interactions with p62 (Figure 5H). The same may be true for various other interaction partners. Therefore, the observed defects may be due to reduced protein-protein interactions or even to reduced membrane tethering/fusion activities in which the N-termini has also been implicated. The authors should therefore attempt to identify point mutation in the N-termini of the ATG8 proteins, which lose cis membrane interactions and test those in cells. This would give greater confidence that loss of cis membrane interaction is indeed the cause of the autophagy defects observed for the deletion mutants.

We thank the reviewer for this helpful suggestion. We introduced three point mutations in GABARAP N-terminal region (M1E/K2E/E7A, Nmut) and characterised this mutant in vitro and in cells. As shown in the revised manuscript (new data in Figure 4, Figure 4—figure supplement 1, Figure 4—figure supplement 2), and discussed starting on page 7, these point mutations partially impaired *cis-*membrane association and had no effect on membrane tethering (*trans-*membrane association) caused by membrane bound GABARAP. The Nmut mutant only partially rescued the autophagosome size of Hexa KO cells: the size was less than half of that observed in the cells expressing GABARAP WT (new data in Figure 7). These results suggest that *cis*-membrane association of the ATG8 proteins positively contributes to autophagosome membrane expansion. As pointed out by the reviewer, impaired membrane growth might be caused by reduced protein-protein interaction. In the revised manuscript, we also show that the GABARAP LIR-docking mutant (Y49A/L50A) fully restored the size of the autophagosomes in Hexa KO cells (new data in Figure 7), indicating that ATG8-cargo interaction is not a major driving force for membrane growth.

2. Regarding the lipid composition used for the in vitro assays, the authors should consider including some PS, PI, or PI3P. These are negatively charged lipids and are present in the phagophore. They may therefore influence the orientation of the ATG8 proteins on the membrane, in particular with respect to RRR/RKR regions.

As shown in the data provided to reviewer, we observed that both LC3B and GABARAP display the conformation where the four key regions namely N-terminal, Loop3, Loop6, and C-terminal were interacting with the charged lipids (65% DOPC/20% DOPE/5% DOPS/10% POPI). These results are in accord with our current simulations with POPC where the above-mentioned regions in LC3 and GABARAP were observed to be interacting with uncharged lipids. Thus, the lipid composition of the membrane may not affect the orientation of single lipidated LC3B or GABARAP.

3. The authors should discuss/mention that the MD simulations do of course favor cis vs. trans interactions with the membrane because as far as I understand no transmembrane (with or without ATG8 proteins) was included. Thus, it is possible that trans interactions are preferred over cis interactions once it becomes available.

We thank the reviewer for this suggestion. Our MD simulation reveals a N-terminal *cis-*interaction conformation of single LC3B/GABARAP-PE on the POPC membrane. As we suggested in the manuscript (Discussion, fourth and fifth paragraph, see page 9), the conformation of ATG8 which favours cis- or trans- interaction may be due to their localisation on the phagophore, concave, convex or highly curved edges. Since previous reports suggest that the trans-interaction of lipidated ATG8 proteins contributes to phagophore membrane expansion via their functions in membrane tethering/hemifusion/fusion, the trans event may favour highly positive membrane curvature, that is, the edge of the phagophore membrane (Nakatogawa et al., Cell, 2007; Weidberg et al., Dev Cell., 2011).

Reviewer #2 (Recommendations for the authors):Atg8 proteins play a dual role in autophagosome formation. On one hand, Atg8s interact with cargo receptors to facilitate cargo recruitment into the inner space of the forming autophagosome. And on the other hand, these proteins were also implicated in the membrane expansion of the phagophore. The exact mechanism responsible for the later activity is still largely unclear. Using liposome assays it has been demonstrated by different research groups, that upon conjugation to PE on the membrane, Atg8s mediate membrane tethering and fusion. However, this view has been challenged as membrane fusion of lipidated Atg8 may require a large amount of PE or a large amount of Atg8 per membrane.In the present study, Zhang et al. utilized a liposome-based assay to determine the mechanism by which GABARAP and LC3B, two members of the Atg8 family, perturbate the membrane upon conjugation to PE. The authors used NBD-labeled Atg8s at different sites to follow changes in NBD fluorescence upon conjugation to the membrane. Conjugation in this system was mediated either enzymatically by the conjugation machinery in the presence of ATP, or chemically, using His6 C-terminus Atg8s and Ni-NTA lipids. Using these systems, the authors found that NBD placed at the N-terminus of the protein indeed changes its fluorescence upon conjugation. The authors also utilized yet another cell-free assay that is based on FRET to measure the interaction between conjugated GABARAP or LC3B with the lipids. This system too indicated a direct interaction between the proteins' N-terminus and the membranes. Finally, the authors utilized the ability of their different GABARAP/LC3B mutants to overcome autophagy inhibition observed upon deletion of all six Atg8 family members. Interestingly, both GABARAP and LC3B recovered the block in autophagy measured by following lysosomal degradation of either p62 or the transfected Atg8 protein. This finding is somewhat surprising as Nguyen et al. JCB 2016 originally indicated that GABARAP proteins but not LC3 are essential for autophagosome formation and targeting of lysosome. More importantly, the authors show that GABARAP mutants lacking the N-terminus 9 amino acids or GABARAP mutated in loop 3 (ARI-3E), maintain the ability to target the mutant Atg8 protein to the lysosome while losing the ability to efficiently engulf p62. A CLEM analysis of the membrane intermediates accumulated by these mutants indicated a defect in their ability to elongate/expand. Based on these findings the authors propose that the "cis" insertion of Atg8 N-terminus to the bilayer is needed for such membrane expansion.Overall, this is a clearly written manuscript and the data obtained from the cell-free systems are straightforward and convincing. The correlation between the cell-free findings and the experiments in tissue culture cells is reasonable, however, they serve as a rather weak support for the suggested model. A more direct in vivo analysis is needed.The main message of this study is the exciting possibility that Atg8s are acting on the membrane in "cis". This is based almost solely, on molecular dynamics simulation. No attempt was made to utilize this method to determine the ability of the different mutants (particularly ARI-3E) used in this study to undergo such conformational changes. Moreover, it is not clear whether the in silico approach took into account the close proximity of another lipid bilayer in "trans". Along these lines, the authors need to directly test whether lipidation of their liposome mediates membrane tethering and fusion, which may imply such "trans" activity.

We thank the reviewer for the helpful comments and suggestions.

Based on NBD fluorescence changes, we clearly showed that GABARAP ARI-3E mutant has a distinct conformation compared to GABARAP WT and that its N-terminus is detached from the membranes in vitro (Figure 5D). However, it would be technically challenging to investigate this conformational change of ARI-3E mutant in vivo. To address this issue, as an alternative approach, we made another *cis*-membrane association mutant, Nmut (M1E/K2E/E7A). In the revised manuscript (new data in Figure 7, see page 8), we found that both *cis*-membrane binding mutants (Nmut and ARI-3E) did not fully restore the size of autophagosomes. Furthermore, we investigated membrane tethering caused by “*trans*” activity of GABARP Nmut, compared to GABARAP WT in vitro (new data in Figure 4, Figure 4—figure supplement 1, Figure 4—figure supplement 2). We found that this impaired *cis*-membrane association mutant GABARAP Nmut had similar “trans” activity as WT protein.

It should be noted that the idea that Atg8 can perturbate the membrane has been recently proposed by Maruyama et al. as indicated by the authors. However, while Maruyama et al. suggest that residues F77 and F79 on the yeast Atg8 protein are inserted into the lipid bilayer (based on NMR studies and GUVs) the current study indicates that the N-terminus is responsible for such perturbation. In both cases, the main conclusions are based on data obtained from cell-free systems, which may turn misleading.

As mentioned above, it is technically difficult and complicated to characterise the conformation of lipidated LC3B/GABARAP in cells. In this study, we aimed to address the molecular mechanism of autophagosome membrane expansion by ATG8 proteins. Maruyama et al. showed that two phenylalanine residues of yeast Atg8 interacts with membranes in vitro, which becomes a driving force for membrane growth of autophagosome. Our model proposed an alternative orientation of lipidated LC3B/GABARAP on the membrane, in which their N-termini associates with the same membrane. Such orientation differences may result from membrane curvature or multiple lipidated LC3B/GABARAPs on the membrane, inducing conformation rearrangement.

In the revised manuscript, we showed that LIR-docking mutant (Y49A/L50A) of GABARAP fully restored the size of autophagosomes in Hexa KO cells. Therefore, we propose that ATG8-membrane association, rather than ATG8-cargo interaction, is crucial for membrane growth, though we cannot address the conformation of ATG8 proteins in vivo. Further analyses are needed in future studies.

The CLEM analysis shown in figure 6 is very impressive, showing that indeed GABARAP is needed for autophagosome expansion. While an important finding, this by itself is not completely new as these proteins were previously implicated in the growth of the autophagic membrane. As indicated above, the conclusion that this is mediated by "cis" acting Atg8 is not demonstrated by the current data. A more rigorous experimental approach aiming to address this directly is needed to solidify this model.

In the revised manuscript, we performed the 3D-CLEM analysis using the Hexa KO cells expressing non-tagged GABARAPs. As far as we know, there is no previous report showing 3D-CLEM image of autophagosomes and quantitative data to assess the requirement of GABARAP N-terminus. To avoid the secondary effects of GFP-tag at the N-terminus of GABARAP, we used non-tagged GABARAPs and analysed membrane growth of autophagosomes. With this approach, we found that *cis*-membrane associated mutants (Nmut and ARI-3E) partially impaired autophagosome membrane expansion, and were significantly different compared to GABARAP WT. Given that N-terminal tagged ATG8 proteins were mainly used in previous studies, we believe that our data is important and informative as new experimental evidence in which the functional requirement of GABARAP N-terminus has been carefully and comprehensively analysed.

The authors' proposal that insertion of the GABARAP/LC3B N-terminus to the membrane will by itself lead to membrane expansion assumes a large ratio of Atg8 to lipid on autophagosomes. This has never been shown and the current study does not address this as well. Alternatively, an exciting possibility may be a link between these proteins on the autophagosomal membrane and Atg2 which was recently implicated in lipid transfer from the ER to the growing autophagosome. The authors are encouraged to test this directly.

Thank you for the suggestions, we agree it is an exciting possibility. To address this, we performed immunoprecipitation experiments (see Figure 6F), but we could not clearly detect the interaction of FLAG-ATG2A with non-tagged GABARAP. Presumably, this interaction was too weak to be detected in our experimental condition. Therefore, we cannot further assess this point.

As for the data shown in figure 5, changes in the p62 level may also result from changes in transcription. The authors should test this directly by RT-PCR or any other method. The IF analysis shown in panel F is missing the Baf. A control. Labeling the intermediates with additional markers such as WIPI, Atg2, etc. could help clarify their nature.

In the revised manuscript, we analysed Hexa KO cells expressing non-tagged GABARAP and found that both *cis*-membrane association mutants (Nmut and ARI-3E) and N-terminal deletion mutants were able to efficiently restore p62-body degradation, indicating that N-terminal region can be dispensable for p62-body degradation (revised new Figure 6). These results are discussed starting on page 7. We have not tested the possibility of a transcriptional regulation as the changes in level could be recovered by BafA1 treatment so we attribute the changes to autophagic degradation. We have revised panel F, now panel D in revised Figure 6, we show the reduction in p62 under starvation to demonstrate the degradation of p62-bodies under starvation does not require *cis*-membrane association.

Reviewer #3 (Recommendations for the authors):The authors examined an important topic, the membrane interactions of the mammalian ATG8 family proteins LC3B and GABARAP, using a combination of molecular dynamics simulations and site-specific NDB labeling, and they find that the N-terminal region of ATG8s binds tightly to membranes. This paper follows the Maruyama et al. study of yeast Atg8 using NMR that came to a different conclusion, that a patch near F77/79 (F80 in human LC3B), binds membranes. The present study concludes that F80 is far from the membrane. The discrepancy may be related to the authors' use of an uncharged lipid mixture in the present study, compared to the more physiological mixture containing 10% PI used by Maruyama. Further, Maruyama used Atg8 covalently conjugated by the physiological conjugation system, while the present study uses a hexahistidine tag to non-covalently tether LC3B and GABARAP to Ni-NTA-containing membranes. Apart from these major limitations in study design, the manuscript generally uses state-of-the-art experimental approaches, and a lot of work clearly went into it, thus it is quite unfortunate that non-optimal choices made in the initial design of the study have rendered the conclusions non-physiologically relevant, or at least, unconvincing.

We agree that Maruyama’s study is relevant and important. To summarize the findings of Maruyama et al. we point out the following: they reconstituted lipidated Atg8 on nanodiscs containing 30% POPE/70%POPC (Methods; Extended Data Figure 2 and Extended Data Figure 3), which is a neutral charge lipid composition. They incorporated 10% PI only in the experiments with GUVs, aiming to facilitate Atg8 lipidation on the GUVs. Meanwhile, after characterisation of residues that interact with lipid nanodiscs via NMR, the authors built up a structure of lipidated Atg8 on POPC membrane surface with HADDOCK. Based on their model on POPC membrane, the authors found that two aromatic residues F77/F79 in Atg8 are close to the membrane and they proposed that these may insert into outer layer of membrane and may induce membrane deformation (referred to Maruyama et al. (2022) “Section: Two aromatic residues in Atg8–PE expand membrane area”, first paragraph).

In our simulation model, we found that the orientation of the side chain in the corresponding residues (LC3B F80, GABARAP F77/F79) are facing towards membrane. However, we did not observe a close contact to the membrane. We found that in our MD simulation, an alternative orientation or conformation of lipidated LC3B/GABARAP, but this does not conflict with Maruyama’s previous model, as Atg8/LC3B/GABARAP may adapt to different conformation due to membrane curvature, membrane geometry or other interactions on the membrane.

In our study, we aimed to start with the simplest lipid model containing only PC or PC/PE and tried to recapitulate, at least in part, the function of membrane association regions in lipidated ATG8, with MD simulation and in vitro assays. We aimed to develop this from simple lipid composition and developed a hypothesis using the POPC membrane system with single ATG8-PE molecules, of how lipidated ATG8 would behave or interact with the membrane. The lipid composition or interactor proteins on autophagic membrane in cells or “physiological” conditions could be very complicated and we did not exclude the *trans* interaction based on previous in vitro studies. In addition, we are proposing an alternative model that would indicate the how the function of lipidated ATG8 distributed on the whole phagophore membrane facilitates membrane expansion.

1. The use of a charge-neutral liposome mixture in both experiments and simulations does not model the phagophore in a realistic way and likely imparts a strong bias to the results. This is the most serious flaw in the study and very likely accounts for the discrepancy with Maruyama. The simulations and NBD interactions should be redone with 10% PI (or some similar charged composition based on knowledge of the mammalian phagophore lipid composition).

We appreciate reviewer’s opinions and suggestions on the lipid composition. We are starting with the simple or basal lipid composition to allow us to investigate more complex lipid compositions or special lipid effects with our approach. Importantly, the basal conditions which has been employed in experiments testing in vitro ATG8 are the conditions which were required when we first developed these assays. We acknowledge the effects of the diverse lipid composition in cell membranes on the conformation of lipidated ATG8. We included experiments with physiological-relevant lipid composition containing 10% PI and reduced PE level to 25%, which has a similar PE level in the ER. As shown in the revised manuscript Figure 4 and Figure 4—figure supplement 2, we observed the membrane insertion of GABARAP N-terminus irrespective of the presence of PI.

MD simulation has been performed in a lipid composition mimicking autophagosome lipid composition (unpublished data). We observed that the orientation of LC3B-PE and GABARAP-PE on the autophagic membrane is similar as that of LC3B-PE/GABARAP-PE on the POPC membrane. The lipid composition of lipid bilayer may not affect the conformation of single LC3B-PE/GABARAP-PE. However, autophagic membranes possess multiple lipidated ATG8 and ATG8-PE can oligomerise (Nakatogawa et al., Cell, 2007). How lipid composition would affect the conformation of multiple ATG8-PE on membrane or whether multiple ATG8-PE would associate with each other on the membrane leading to a conformational rearrangement requires further investigation.

2. The use of the C-terminal His6 tag construct in the experimental setup was an ill-advised choice in the experimental design, and may substantially perturb the results. The authors should perform the correct conjugation experiment with LC3B(G120) and GABARAP (G116) and redo all of the NDB probe experiments in this condition. The physical chemistry of the His tag linkage is very different from that of the normal amide linkage of the C-terminal Gly to the PE headgroup. The introduction of a nickel ion potentially confounds the measurement of the membrane associations. In this respect, the experiments do not match the simulations, so direct comparisons cannot be made.

In our study, to access the lipidation activity of LC3B/GABARAP, we purified the glycine-exposed protein LC3B (G120) and GABARAP (G116) in the real-time lipidation assay using NBD probe. As shown in our manuscript Figure 1, we were able to distinguish the conjugation steps of LC3B/GABARAP to ATG7, ATG3 or finally PE. To access the protein-membrane association mimicking the lipidated state of LC3B/GABARAP, we used the C-terminal His_6_-tagged LC3B/GABARAP with NBD labelling on the specific residues.

So far, there is only two chemically conjugation system available to mimic ATG8-PE, one is maleimidethiol coupling (cysteine mutation at LC3B G120C or GABARAP G116C) and PE-maleimide, the other one is poly-histidine and Ni-NTA. Since NBD fluorescence conjugation need Cysteine mutation, the maleimide-thiol coupling is not adaptable in our NBD fluorescence system. ATG8-his has been employed in previous studies to coupling C-terminus of ATG8 to a membrane and mimic the membrane bound ATG8 (Sawa-Makarska et al., Nat Cell Biol., 2014; Taniguchi et al., Protein Science, 2020).

The His-tag on the ATG8 C-terminus is one of the limitations in our experiment design. We agree that it cannot fully represent the covalent conjugation between ATG8 and PE, while there are advantages employing polyhistidine and nickel lipids to investigate ATG8-membrane interaction. Because of the strong affinity between polyhistidine and nickel lipids, the readout of our in vitro assay directly reflect the membrane tethering activity or *cis-*membrane association of GABARAP (post-lipidation).

Additionally, in our case, we would like to have a reversible conjugation system by adding imidazole to remove LC3B/GABARAP from liposomes, so we can confirm the protein-membrane interaction (FRET assay).

Our in vitro results match our MD simulation mostly, except for the RKR regions in GABARAP. As we explained in the manuscript, this discrepancy between MD simulation and FRET assay on GABARAP RKR region may result from the conformational rearrangement of multiple GABARAP conjugated on the single liposomes in the FRET assay, whereas the MD simulation demonstrates the single GABARAPPE on the lipid bilayer.

3. The authors advance the idea that the membrane association of an amphipathic helix of ATG8's Nterminus is required for autophagosomal expansion in the case of P62 aggregates. The basic residues of the first alpha helix of LC3B interact with the acidic residues of p62's LIR domain. The authors' data shows that pull down with p62 is weakened by the loss of the N-terminal helix. A necessary control experiment, in this case, is for a measurement of the drop in affinity of p62 for LC3B and GABARAP, as the reduction of association between ATG8 and the cargo receptors could present an alternative explanation for the reduction in autophagosomal expansion efficiency for the p62 clearance case.

ATG8 N-terminal helix is not an amphipathic helix.We thank the reviewer for the helpful suggestions. In the revised manuscript, we showed that LIRdocking mutant (Y49A/L50A) of GABARAP fully restored the size of autophagosomes in Hexa KO cells, indicating that ATG8-cargo interaction is not a driving force for membrane growth of autophagosomes (revised Figure 7 and Figure supplemental 1)